# HUMANLM: Simulating Users with State Alignment Beats Response Imitation

**Shirley Wu** [* 1]  **Evelyn Choi** [* 1]  **Arpandeep Khatua** [* 1]  **Zhanghan Wang** [2]  **Joy He-Yueya** [1]
**Tharindu Cyril Weerasooriya** [3]  **Wei Wei** [3]  **Diyi Yang** [1]  **Jure Leskovec** [** 1]  **James Zou** [** 1]

[*]*Equal Contribution*    [**]*Equal Senior Supervision*

http://humanlm.stanford.edu/

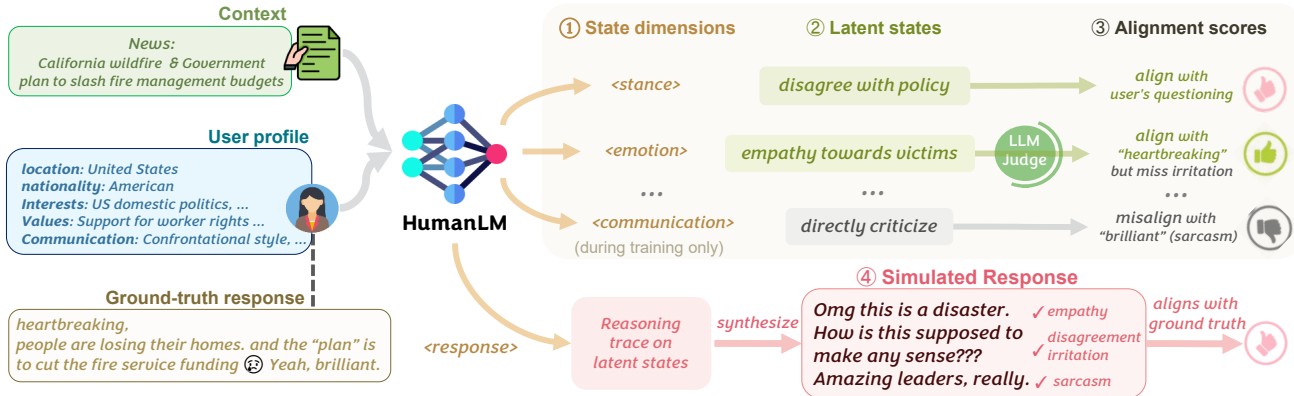

*Figure 1.* HUMANLM generates responses that capture the key points of real user responses. Given an input context (*e.g.,* a news post) and a user profile, the model prioritizes alignment along a few psychologically grounded ① state dimensions (*e.g.,* stance, emotion), that lead to how users respond. For each state dimension, the model generates the corresponding ② latent state (*e.g.,* "empathy toward victims"), ③ scored by an LLM judge for consistency with the ground-truth response. During reinforcement learning, the model maximizes alignment scores on latent states to accurately reflect real users, in addition to directly improving the responses. When generating responses, the model generates reasoning traces with aligned latent states to synthesize ④ accurate responses.

## Abstract

Large Language Models (LLMs) are increasingly used to simulate how specific users respond to any context, enabling more user-centric applications that rely on user feedback. However, existing user simulators mostly imitate surface-level patterns and language styles, which fails to reflect the underlying state of real users (*e.g.,* beliefs, emotions). To address these limitations, we propose a novel training framework, HUMANLM, which builds user simulators that accurately reflect real users. Our key insight is, in addition to generating responses, we generate natural-language *latent states* that align with the ground truth responses through reinforcement learning. These latent states correspond to a set of *state dimen-*

*sion*s which psychologically lead to how real users respond. HUMANLM further synthesizes these aligned latent states into responses that accurately represent real users. For extensive evaluation, we develop HUMANUAL, a comprehensive benchmark on simulating real users based on public data. HUMANUAL consists of six large-scale datasets with 23k users and 227k responses in total. It spans diverse tasks such as generating user responses to daily life issues, political blogs, and chat sessions with LLM assistants. Across the datasets, HUMANLM significantly outperforms the best alternative approaches by an average relative improvement of 16.3% on alignment score from an LLM judge. In a real-time simulation study with 111 participants, HUMANLM achieves the highest scores on similarity with real user responses and humanlikeness.

[1]Stanford [2]New York University [3]Accenture. Correspondence to: <{shirwu,jure,jamesz}@cs.stanford.edu>.

*Proceedings of the 43rd International Conference on Machine Learning*, Seoul, South Korea. PMLR 306, 2026. Copyright 2026 by the author(s).

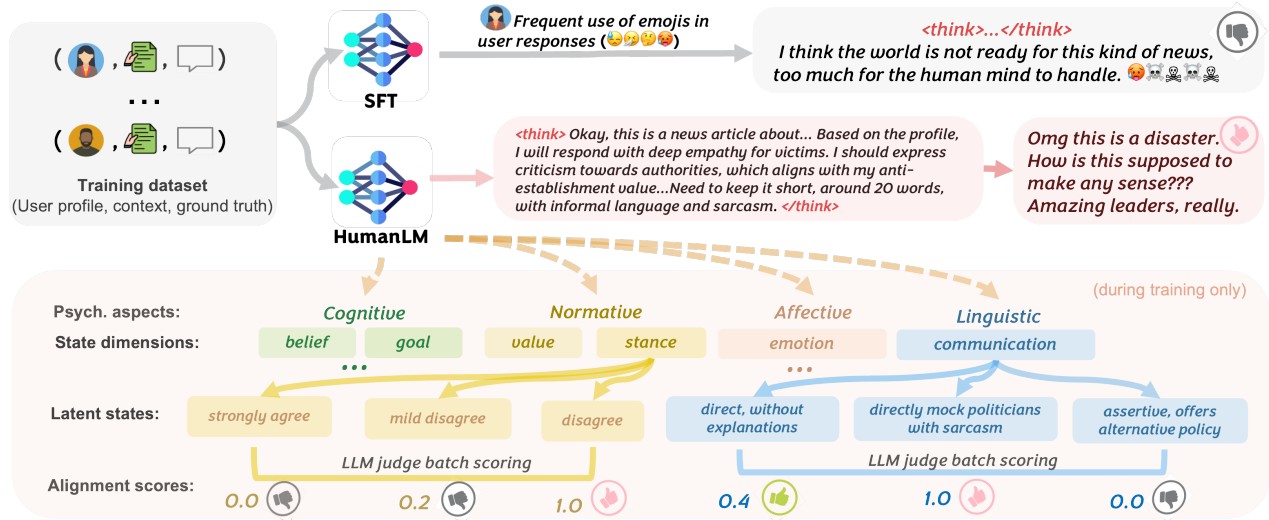

*Figure 2.* Comparison between HUMANLM and Supervised Fine-Tuning (SFT). Given a training dataset, SFT learns to capture the frequent use of emojis of the user, resulting in an inaccurate response that misses the key points in the ground-truth response (*cf.* Figure 1) during evaluation. In contrast, HUMANLM explicitly learns to align along different state dimensions, generating latent states that reflect the user in the reasoning trace, which leads to a more accurate response. We apply GRPO (Shao et al., 2024) for reinforcement learning, where an LLM judge is prompted to compare a batch of generated latent states under each state dimension (*aka.* rollouts) and give alignment scores for them at once, providing more precise rewards under fair comparisons.

# 1. Introduction

Simulating users using Large Language Models (LLMs) helps to understand how a target user group will respond to any input context, providing a scalable way to build human-centric services and applications (Binz et al., 2025; Naous et al., 2025; Kolluri et al., 2025; Park et al., 2022). For example, policymakers, writers, and AI model developers can leverage responses from user simulators to improve policies, articles, and AI features to receive target outcomes (Wu et al., 2025; Hwang et al., 2025; Qian et al., 2025b; He-Yueya et al., 2024). However, existing LLM-based user simulators are primarily trained to imitate surface-level language use in user responses, instead of capturing higher-level user states, such as user stance to support a policy, emotions to favor an AI response, or values in evaluating articles, which drive real-world outcomes (Chuang et al., 2025; Lu et al., 2025; Kolluri et al., 2025; Binz et al., 2025; Naous et al., 2025). As a result, current user simulators provide unreliable user responses that do not reflect real user behaviors.

Here we present **HUMANLM**, a novel framework to train LLM-based user simulators that capture the underlying states of users. Our key insight (Figure 1) is to align a model with multiple *state dimensions* that drive user responses. These state dimensions, such as stance and emotion, provide axes for the model to generate a set of specific *latent states*, such as "*disagree with the policy*" (stance) or "*empathy towards victims*" (emotion). By fine-tuning with RL algorithms (Shao et al., 2024) to maximize alignment scores on these latent states, which measure if each

latent state is consistent with the ground truth response, the model prioritizes learning higher-level user states that reflect real user properties. When prompted for responses under unseen contexts, HUMANLM generates reasoning traces with aligned latent states and further synthesizes responses. Figure 2 shows a reasoning trace where HUMANLM accurately captures multiple states. Compared to text imitation, HUMANLM's response contains more similar key points expressed in the ground truth.

To evaluate user simulators, we introduce **HUMANUAL** (Figure 3), a comprehensive benchmark in simulating user responses. Existing user simulation benchmarks usually rely on simplified or synthetic user profiles (Castricato et al., 2025; Kirk et al., 2024; Kumar et al., 2025) and limited context scopes (Binz et al., 2025; Santurkar et al., 2023). In contrast, HUMANUAL comprises six datasets from publicly available sources with rich, real user profiles, including Reddit users discussing life issues, Medium users giving blog feedback, and Amazon users reviewing books (He & McAuley, 2016). In total, HUMANUAL spans over 23k users worldwide and 227k diverse responses on 67k topics. Across the datasets, HUMANLM substantially outperforms prior approaches with prompting, supervised fine-tuning, and reinforcement learning by 16.3%.

Moreover, we conduct **real-time simulation** with 111 participants. Each participant responds to a randomly sampled Reddit post and compares their response with the simulated responses from three different models. Upon finishing, they rate the overall similarity and humanlikeness of each simulated response on a scale from 1 to 10. Among three

user simulators, HUMANLM achieves the highest win rate of 41.4% on overall similarity: 55.9% of participants rate HUMANLM responses as "*mostly similar*" or "*nearly identical*" to their own, compared to only 45.0% for the best baseline. HUMANLM also generates more natural-sounding responses, with 76.6% of responses above "*quite natural*".

## 2. Problem Formulation

We consider a generic dataset $\{(p^{(i)}, x^{(i)}, y^{(i)})\}_{i=1}^N$. Here, $p$ represents a user persona created from any user identifiers such as user profile, IDs, or persona summarized from user history. $x$ is the input context, which can be either single-turn (*e.g.,* news reports, blogs) or multi-turn (*e.g.,* a back-and-forth conversation between user and an LLM assistant, social media posts along with other users' follow-up comments). $y$ is the ground-truth response from the user ($p$) to the input context.

For any input context $x$, we define a **latent state space** $\mathcal{S}(x) = \{s_1, s_2, \ldots\}$ with a finite number of latent states. Each latent state represents a distinct high-level attribute that a response may express or reflect, such as "*deep heartbreak for the wildfire victims*", "*irritation about the government's untimely rescue*", and "*provide claims with evidence*". [1]

For an arbitrary response $y$, we define a mapping $M : y \rightarrow \{s_{j_1}, s_{j_2}, \ldots\}$, where each index $j_i \in [|\mathcal{S}(x)|]$. For any input context $x$, our goal is to generate response $\hat{y}$ such that the latent states from the generated response match those from the ground truth

$$\min_{\hat{y}} \sum_{j=1}^{|\mathcal{S}(x)|} \left| \mathbb{I}(s_j \in M(\hat{y})) - \mathbb{I}(s_j \in M(y)) \right|, \quad (1)$$

where $\mathbb{I}$ is an indicator function. The above formulation regards a response as a bag of latent states, where the objective penalizes missing latent states or redundant latent states outside of the ground-truth responses.

## 3. Training Aligned User Simulators

**Motivation**. Previous works optimize the objective by training models to imitate the exact ground-truth responses (Naous et al., 2025; Binz et al., 2025). Note that when a generated response $\hat{y}$ exactly matches the ground-truth $y$, the objective in Eq. 1 achieves a lower bound.

However, imitating ground-truth responses is often infeasible in practice, since user responses are non-deterministic by nature. In fact, even the same user may not perfectly reproduce their own responses. For example, a user may choose to use different phrases like "*not a good start*" or

"*bad idea*" to express the same stance of disagreement.

Moreover, this focus on surface-level language can easily prevent models from learning meaningful latent states. For example, a user may convey disagreement through sarcasm ("*well, what a promising start*") or through straightforward criticism ("*bad idea*") with emojis. Here, imitating specific language use (*e.g.,* a more frequent use of emojis and negative words like "*bad*") may fail to capture the user's high-level communication behavior (*e.g.,* sarcasm *v.s.* directness), thus mismatching with ground truth when given unseen contexts. Therefore, instead of imitating ground-truth responses, our focus is to align model generations with latent states inferred from ground-truth user responses.

### 3.1. From Post-hoc to Ad-hoc Alignment

**Challenges**. A straightforward solution for latent state alignment is to reward a generated response by how much it aligns with the ground truth in terms of the latent states, referred to as **response alignment scores**. For a given context, we can prompt an LLM judge to 1) extract the key latent states for a generated response and ground-truth response separately and 2) compute the match score between these two sets of latent states. We can then apply reinforcement learning (RL) algorithms such as GRPO (Shao et al., 2024) to optimize the model for higher response alignment scores.

However, since we aggregate all latent state matches, it is unclear which underlying latent states were correct or incorrect during reward assignment. For example, consider a real user response in Figure 1, which conveys multiple latent states: empathy towards victims, disagreement with the policy, and use of sarcastic criticism. In this example, generated responses that match any one of the latent states and mismatch on the others can achieve similar rewards. As a result, it creates **combinatorial ambiguity** during training, which "confuses" the model about which latent states should be improved and how to improve them.

**Key idea**. Built on the insights, our idea is to explicitly generate latent states and treat responses as outcomes conditioned on latent states, rather than as the source from which latent states are inferred. This reframes the problem. Instead of asking "given the extracted latent states, is this response well-aligned?", we ask "how can we generate aligned latent states such that given these states, the synthesized responses are aligned?". Therefore, we decompose the problem into (Section 3.2) generating aligned latent states and (Section 3.3) synthesizing latent states into responses. Finally, Section 3.4 provides a full picture of our method.

### 3.2. Generating Aligned Latent States

We train a user simulator to generate multiple latent states. Our idea is to design *state dimensions* (*i.e.,* axes for latent

---

[1]Formally, let $\text{sim} : \mathcal{S}(x) \times \mathcal{S}(x) \rightarrow [0, 1]$ be a similarity function and let $\tau \in (0, 1)$ be a granularity threshold. We define states to be distinct only if $\forall s \neq s', \text{sim}(s, s') \leq \tau$.

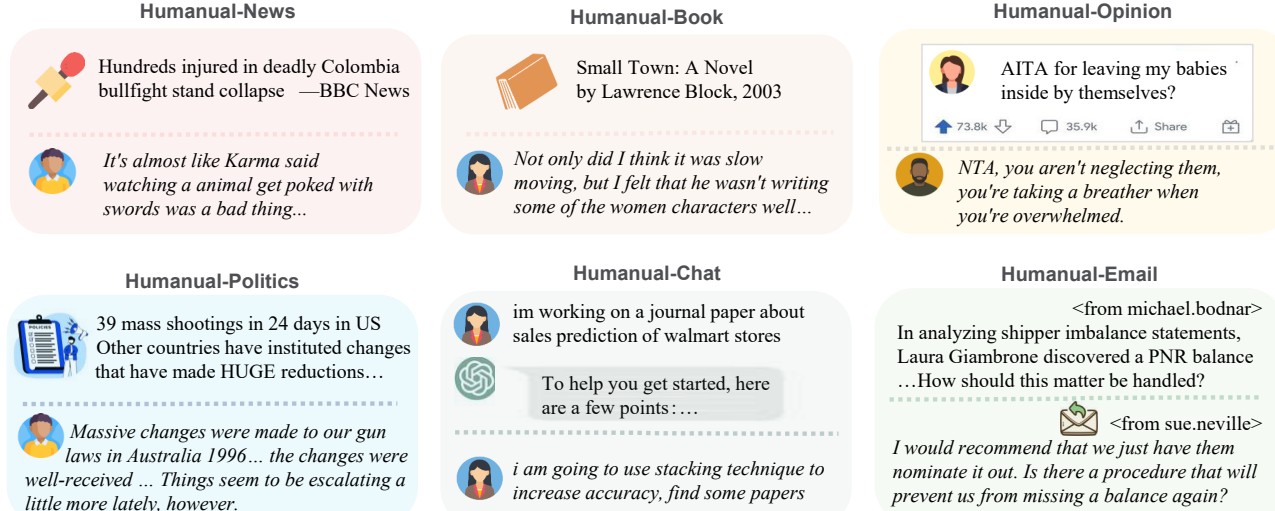

*Figure 3.* Examples (context ⦂ *ground truth*) from HUMANUAL, which covers six diverse domains including simulating news comments, book reviews, opinions on daily life issues, political blogs, email replies, and follow-ups with LLM assistants.

state values), to capture how people think, take positions, and express themselves, which jointly form the responses.

**State dimensions:** belief, goal, emotion, value, stance, and communication are motivated by four psychological aspects:

- **Cognitive aspect** (*belief*, *goal*) is based on the Belief–Desire–Intention framework (Rao & Georgeff, 1991). Beliefs describe what a user thinks is true, while goals describe what the user wants to achieve.

- **Normative aspect** (*value*, *stance*) distinguishes between what users care about and their position in a specific social context, drawing from sociolinguistics and positioning theory (Davies & Harré, 1990). A user who values honesty may still tell a child that Santa Claus is real.

- **Affective aspect** (*emotion*) is a short-term process that changes how information is acted upon (Zajonc, 1980; Sander et al., 2005). As a result, two users can have the same stance (disagreement with a policy) but radically different emotions (outrage *v.s.* worried).

- **Linguistic aspect** (*communication*) captures how information is expressed (Levelt, 1989). Different from surface-level language use, we refer to communication as the way users structure their responses: whether they respond directly or indirectly, assert claims or provide evidence, give answers or ask questions, *etc.* Responses that differ in communication can lead to distinct interactions.

While some state dimensions may be weakly expressed in responses, they are generally present in the underlying response generation process (Levelt, 1989).

**Alignment scores on latent states**. The state dimensions provide basis for latent state alignment. In each training batch, we randomly sample one state dimension and prompt

the user simulator to generate the multiple corresponding latent states. We then use an LLM judge to score (from 0-1) on how consistent the generated latent states are with the ground truth response along that state dimension.

Yet, assigning a score one at a time with an LLM judge introduces significant bias due to the lack of comparison. For example, the LLM judge may assign the same score of 1.0 to two latent states about communication, "*direct, without explanation*" and "*directly mock politicians with sarcasm,*" when evaluated separately, even though the latter is more comprehensive and accurate. To avoid bias score assignment, in Figure 2 we sample a batch of latent states for the same context (*i.e.,* rollouts) and prompt the LLM judge to score them comparatively. Later, these scores are used as rewards in the model's training process (Section 3.4), reinforcing the model to generate aligned latent states under the state dimensions.

### 3.3. Synthesizing Responses from Aligned Latent States

Each latent state may not contribute equally to a response. In fact, some latent states may overlap in content. As a result, simply summarizing all of the generated latent states can introduce redundancy or even inconsistency. Both cases undermine the objective in Eq. 1. Moreover, human language production integrates multiple interacting constraints into a single utterance through unification, rather than expressing each factor independently (Hagoort, 2013; Pessoa, 2008). This motivates a synthesis process to model multiple latent states into the final response.

**Response synthesis**. We prompt the model to generate reasoning traces with user latent states. Later in the experiment section, we validate that these reasoning traces include latent states learned from explicit latent state alignment.

Moreover, in the reasoning traces, the model also analyzes how these latent states impact the final response, such as how to organize it (*e.g.,* "*start with deep empathy*"), which latent states to emphasize, and which to make more concise, *etc.* Based on these intermediate rationales, the model generates responses consistent with the latent states. We compute response alignment scores (*cf.* Section 3.1) on the generated responses using an LLM judge.

### 3.4. Training and Inference

In Figure 1, given the ① state dimensions, we train a user simulator to generate the corresponding ② latent states. When prompted for a full response, the user simulator first generates a reasoning trace that reasons about these latent states, and then synthesizes ③ the final response. We use an LLM judge to compute ④ alignment scores for *both* the generated latent states and the generated responses in a batch (Figure 2), where outputs/rollouts with the same inputs are evaluated under comparison. We use these scores as rewards for reinforcement learning (RL), such as GRPO (Shao et al., 2024). In training, we prompt the user simulator to generate a batch of outputs with mixed latent states and responses. In testing, we only prompt the user simulator to generate responses with reasoning traces and evaluate using the generated responses.

## 4. Benchmark and Experiment Setup

**Benchmark (Figure 3)**[2] We create HUMANUAL, a benchmark for user simulators, consisting of six diverse datasets from real and publicly available data sources. We have included additional details in Appendix A. Here, we describe each dataset briefly:

- **Humanual-News** contains comments from 10.9k YouTube users on 6.1k videos posted by BBC and CNN channels, totaling 43k comments. This dataset highlights users' different reactions or targets regarding news events. We use the video transcriptions as the input contexts.

- **Humanual-Book** contains 40k Amazon book reviews from 209 frequent customers, each with 192 reviews on average (He & McAuley, 2016). The reviews express satisfaction or dissatisfaction with book content, reflecting users' preferences and tastes.

- **Humanual-Opinion** contains 4.6k Reddit users expressing opinions across 1k diverse personal-issue threads, resulting in around 46k responses. These responses reflect users' moral standards on controversial topics, *e.g.,* family conflicts and life decisions.

- **Humanual-Politics** consists of 5.3k Medium users and 50k responses in total to 15k blog posts on political topics. It features diverse political stances from real users

spanning different cultural backgrounds, and is intended to simulate user responses to long-form written content.

- **Humanual-Chat** consists of conversations between users and LLM assistants of 5–10 turns, adapted from WildChat (Zhao et al., 2024). The goal is to simulate interactive user behaviors with LLM assistants, including follow-ups, goal changes, and clarification turns.

- **Humanual-Email** has 399 users and 5.2k email threads, adapted from the Enron email dataset (Cohen & CALO Project, 2015). It captures user communication in business settings, including decision negotiation, project status reporting, and constraint resolution.

**Official data splits**. For `Humanual-Chat`, we split by turns within each conversation, assigning the earliest 80% of turns to the training set. For the other datasets, we arrange original contexts (*e.g.,* posts, news, blogs) by timestamp and divide contexts into different splits chronologically; therefore, the test contexts are **unseen** in the training datasets. All processing steps are made transparent in our code.

**User profile** (*cf.* Appendix G.1 for prompts). For datasets except `Humanual-Chat`, we summarize a user profile for each user from at most their earliest 20 responses in the train set using `claude-4.5-haiku (20251001)`. The user profiles cover potential demographics, interests, and communication examples *etc.* We do not construct profiles on `Humanual-Chat` due to a lack of precise user identifiers.

**Evaluation metrics** (Appendix G.2). For each generation, we prompt an LLM judge to give a **response alignment score** consistent with Eq. 1. For the quality of latent state alignment, we compute **state alignment scores** by prompting the LLM judge to evaluate how well model generations align with the ground-truth responses along one of the six state dimensions. We use `claude-4.5-haiku` as the judge model (see the Appendix G.2 for prompts). To provide a more deterministic evaluation, we compute the cosine similarity between generation and ground truth embeddings (see Appendix D for the analysis).

**Baselines** (Appendix B). HUMANLMs are trained from Qwen3-8b, compared to seven baselines:

- **Qwen3-8b**, the base model, and **Qwen3-8b-think** with step-by-step reasoning before generating responses;

- **SFT**: Supervised fine-tuned models trained to imitate ground-truth responses;

- **SFT-think** (Lu et al., 2025): We generate synthetic user thoughts that lead to the ground-truth responses by prompting `gpt-5-mini`. Then, we conduct SFT on these synthetic thoughts with the ground-truth responses.

- **UserLM** (Naous et al., 2025): A model post-trained on WildChat (Zhao et al., 2024) from Llama3-8b-Base to simulate users in multiturn. Applicable only for the `Humanual-Chat` benchmark.

---

[2]Samples of data and user profiles in the anonymous website.

*Table 1.* **Response alignment scores** (↑) on HUMANUAL. Last row shows HUMANLM's relative improvements to the best baselines.

| | News | Book | Opinion | Politics | Chat | Email | Avg. |
|---|---|---|---|---|---|---|---|
| Qwen3-8b | 5.68 | 13.6 | 18.7 | 10.1 | 3.90 | 4.76 | 9.5 |
| Qwen3-8b-think | 4.83 | 12.8 | 20.4 | 7.0 | 2.16 | 3.22 | 8.4 |
| SFT | 3.10 | 9.3 | 11.3 | 6.3 | 4.57 | 4.30 | 6.5 |
| SFT-think | 6.00 | 13.4 | 16.7 | 9.2 | 2.50 | 3.94 | 8.6 |
| UserLM | - | - | - | - | 2.47 | - | - |
| GRPO | 7.92 | 13.3 | 18.2 | 10.9 | 5.83 | 5.90 | 10.3 |
| GRPO-think | 7.04 | 12.8 | 23.8 | 10.6 | 3.16 | 4.78 | 10.4 |
| HUMANLM | 9.55 | 18.5 | 25.6 | 12.6 | 6.08 | 6.71 | 13.2 |
| Rel. Improvement | 20.6% | 36.0% | 7.6% | 15.6% | 4.3% | 13.7% | 16.3% |

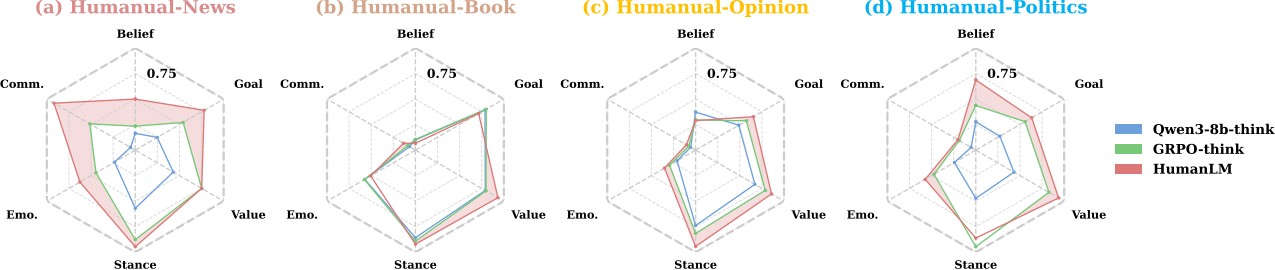

*Figure 4.* **State alignment scores** (↑) of HUMANLM and two baselines on four HUMANUAL datasets. Full results in Appendix D.

- **(Standard) GRPO**, and **(standard) GRPO-think** (Shao et al., 2024): RL-trained models using Group Relative Policy Optimization (GRPO). We directly use the response alignment scores by a judge, `gpt-5-mini`, as rewards; GRPO-think generates reasoning traces before responses.

**HUMANLM Implementation** (Appendix C). We train models on the training sets using the same hyperparameters. Note that we use `gpt-5-mini` as the LLM judge in training, different from the judge (`claude-4.5-haiku`) in testing, to ensure a more reliable and unbiased evaluation.

## 5. Results on Benchmark

We report the main results in Table 1 and Figure 4, with the following conclusions:

**1) Simulating real-world user responses is still an extremely challenging task.** The Qwen3-8b model's average score across the datasets is around 10%, showing that real user responses are hard to simulate due to highly complex user profiles and diverse contexts. As a result, enabling reasoning or learning on high-quality reasoning traces (*e.g.,* SFT *v.s.* SFT-think) lead to improvements on some datasets.

**2) SFT discourages learning meaningful user states.** Through extensive training to predict next tokens on large-scale datasets, SFT-based approaches consistently perform the worst among all methods. Under careful inspection, we find that while SFT generated responses mimic user tones well, they tend to be overly long and frequently hold opposite opinions compared to the ground truth, validating that imitating user responses hardly captures higher-level states.

**3) Directly optimizing alignment scores leads to improvements.** We find that standard GRPO approaches outperform SFT by some margins during testing, while some improvements are marginal, such as 3.94 (SFT-think) → 4.78 (+0.84) (GRPO-think) on `Humanual-Email`.

**4) HUMANLM generates highly aligned responses and states.** Table 1 shows that HUMANLM consistently achieves the best response alignment scores with an average relative improvement of 16.3%. Specifically, HUMANLM achieves relative improvements of 38% and 17% over base-think and GRPO-think, respectively. In Figure 4, our model achieves the highest alignment scores on 80% of the latent states.

**Embedding similarity (Appendix Table 3).** Despite not using this metric as the reward, HUMANLM improves embedding similarity between generated responses and the ground truth by 7.5% compared to Qwen3-8b-think.

**Evaluation reliability check (Figure 6).** To validate that alignment scores are not biased towards a specific judge model, we use another judge, `gemini-3-pro` to evaluate models on `Humanual-Politics`. Figure 6 shows consistent model rankings across judges, with HUMANLM ranked first by both judges.

### 5.1. Training Dynamics of HUMANLM (Figure 5)

We provide insights to explain why HUMANLM generates more aligned responses. Figure 5 compares the training dynamics between HUMANLM and GRPO-think, which both train on response alignment scores. For each method, we compute the average state and response alignment scores

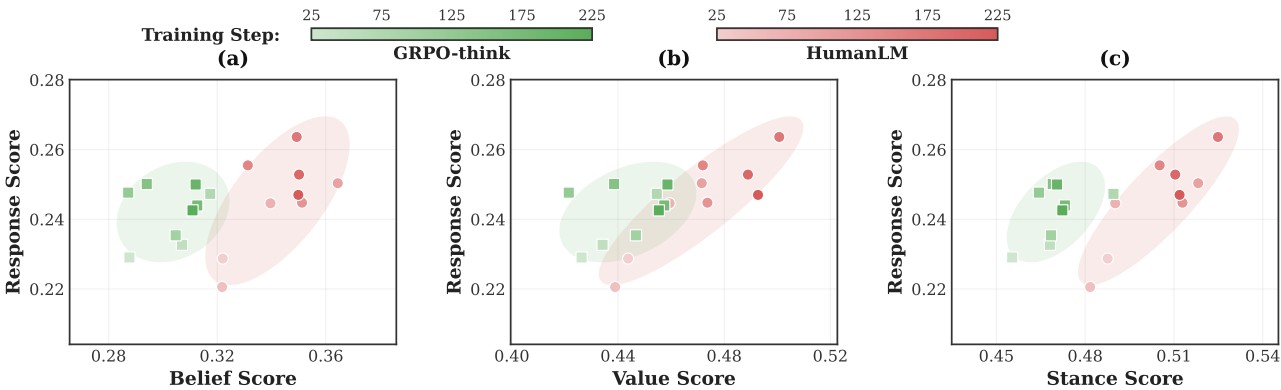

*Figure 5.* **Training dynamics comparison** between HUMANLM and GRPO-think. Each dot represents a model checkpoint saved every 25 steps when training on `Humanual-Opinion`. Each $x$ value is the checkpoint's alignment score along one of the state dimensions: belief, value, and stance. Each $y$ value is the response alignment score. Compared to GRPO, HUMANLM shows broader score coverages through exploring states with explicit alignment, which encourages more optimal alignment on responses. Full results in Appendix D.

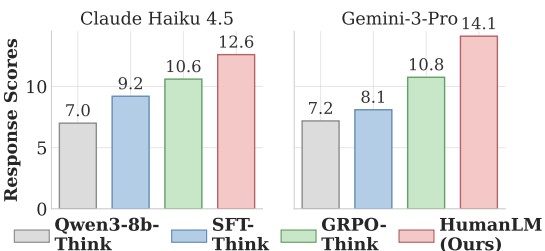

*Figure 6.* **Consistent rankings** from different LLM judges for evaluating response alignment on `Humanual-Politics`.

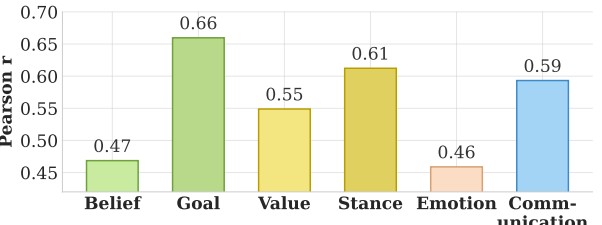

*Figure 7.* **Dimension importance** on `Humanual-Opinion`. Goal and stance scores are largely correlated with response scores.

for multiple checkpoints saved during training, evaluated on 500 validation samples in `Humanual-Opinion`.

We find that GRPO-think yields a highly limited range in state alignment scores during training, indicating that the models are "stuck" and struggle to find consistent directions for exploring each state. This validates our earlier claim that models fail to consistently interpret responses with similar scores but different combinations of latent states. As a result, this leads to limited or inconsistent exploration of responses, undermining alignment quality.

In contrast, HUMANLM yields higher response alignment scores from **consistently exploring** different states. Specifically, HUMANLM shows broader score coverage during training, where the average spans on state and response alignment scores are 23% and 104% higher than GRPO-think, respectively. By explicitly generating latent states, the model receives clear signals to align with latent states in the ground truth. This mitigates local optima when relying only on response alignment scores.

### 5.2. Relations between States & Responses (Figure 7, 8)

We study how different state dimensions contribute differently to responses. To estimate the contribution, we define the **dimension importance** as the Pearson Correlation value between response alignment scores and the state alignment

scores along a state dimension. Figure 7 reports the results based on 1k simulated responses for `Humanual-Opinion`, where goal and stance are among the first tier. This is consistent with the task property, where most users take explicit goal-oriented actions (*e.g.,* give suggestions to poster) and stances (support *v.s.* disapprove).

We further study how reasoning traces with latent states contribute to final responses. We present three case studies in Figure 8, which demonstrates three reasoning traces and the corresponding generated response. The key takeaway is that the reasoning traces broadly include the latent states from all state dimensions, which are well reflected in the final natural-sounding responses. For example, the reasoning trace in Figure 8b involves a stance of "`affirm the user's stance`", a value of "`personal boundaries`", and a communication style of "`concise and empathetic but firm`". These together lead to a final concise response that is supportive of the poster's actions, with reasons emphasizing that others should respect personal boundaries.

## 6. Real-time User Simulation

**Setup**. To evaluate how well HUMANLM generalizes to users with different profiles, we asked 111 Amazon Mechanical Turkers to write down their own responses to a Reddit post sampled from `Humanual-Opinion` test set

**Cognitive aspects**
*belief*   *goal*

**Affective aspect**
*emotion*

*<think> ... might have a critical view towards the involved parties, possibly the Serbian forces, and might use a confrontational tone. Since the persona is critical of US military interventions and uses sarcasm, they might express skepticism or criticism towards those in power. Also, they might use a short, direct response with emotional expression, possibly ... make sure the response is short, around 15-20 words, with informal language.*

*... references to similar historical events and modern parallels, but in this case, ... express a strong emotion. Also, the persona may use a rhetorical question or a statement that highlights hypocrisy or criticism. </think>*

*<response> Same here. Another reminder of how some still deny the truth, while others profit from it. Sad how history repeats with silence. </response>*

**Normative aspects**
*value*   *stance*

**Linguistic aspect**
*communication*

*(a)* `Humanual-News`

**Cognitive aspects**
*belief*   *goal*

**Affective aspect**
*emotion*

*<think> Okay, let's break this down. ... The key points here are the user's need for the chair due to their own health issues and the pregnant woman's entitlement....*

*The response needs to affirm the user's stance while acknowledging the woman's situation. The persona's principles emphasize NTA, supporting personal boundaries, and holding people accountable for entitlement.*

*I should start with NTA to validate the user. Mention their legitimate need for the seat. Point out the woman's assumption of entitlement. Suggest she should have brought her own chair.*

*Maybe add a light-hearted comment about the husband's reaction. Keep it concise and empathetic but firm. </think>*

*<response> NTA. You're not obligated to give up your seat, especially when you brought your own and have legitimate reasons to need it. ... It's reasonable to decline without being rude. Keep your boundaries; you're not required to accommodate others' discomfort at your expense. </response>*

**Normative aspects**
*value*   *stance*

**Linguistic aspect**
*communication*

*(b)* `Humanual-Opinion`

**Cognitive aspects**
*belief*   *goal*

**Affective aspect**
*emotion*

*<think> Okay, let me start ... they value activism, justice ... so they'd likely see the article's points as aligning with their views.*

*... uses direct and confrontational language, often ... could tie into the article's point about persistence against systemic issues despite personal challenges. They might feel a kinship with the article's message of resistance and **solidarity**.*

*...tendency to use humor and sarcasm ... concise, impactful, while adding personal anecdotes or calls to action, aligning with their activism and belief in collective resistance. </think>*

*<response> YES! YES!! That's exactly what I've been saying for year ... And the worst part? It's not just about money – it's about eroding the very idea that public institutions serve everyone.*

*And the narrative? Oh, the narrative! "Forgotten Americans" being victimized by "elites"– but the elites? They're the ones profiting. It's a masterclass in reverse victimhood. And yes – solidarity is the only antidote. </response>*

**Normative aspects**
*value*   *stance*

**Linguistic aspect**
*communication*

*(c)* `Humanual-Politics`

*Figure 8.* Reasoning traces and responses decomposed into six state dimensions. The examples show how the generated latent states in the reasoning traces jointly shape the final responses across real-world domains, such as news, daily-life, and political discussion.

(79 posts) and compare their responses against three simulated responses from one of Qwen3-8b-think, GRPO-think, and HUMANLM. See Appendix H for details.

To generate the user profiles for these user simulators, we ask them to first answer a few open-ended questions and summarize their values and communication styles. After the participants finish their responses, we present three simulated responses in random order. The participants then give overall similarity scores and humanlikeness scores after comparing the simulated responses with their own.

**Results (Figure 9).** For overall similarity scores, HU-MANLM achieves the highest average score of 6.5 with a win rate (*i.e.,* percentage of model responses that receive the highest similarity scores among all three models) of 41.4%. In contrast, Qwen3-8b-think and GRPO-think arrive at win rates of 30.6% (–10.8%) and 27.9% (–13.5%), respectively. 68.6% of the participants reported that HUMANLM responses are "*most similar*" or "*nearly identical*" to theirs.

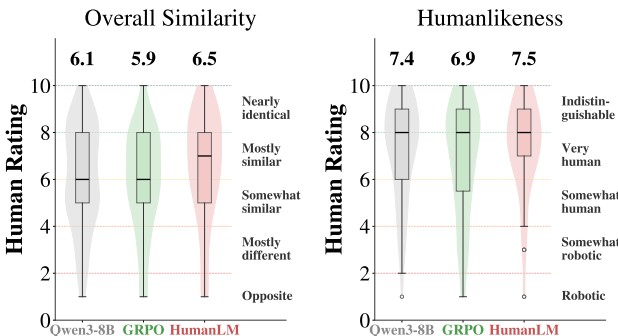

*Figure 9.* **Overall similarity and humanlikeness scores**

**Statistical significance**. We assess whether HUMANLM's improvements in overall similarity are statistically significant. We conduct paired one-sided Wilcoxon signed-rank tests across the scores from 111 participants, confirming that HUMANLM significantly outperforms both Qwen3-8b-think ($p = 0.0279 < 0.05$) and GRPO-think ($p = 0.00284 < 0.01$).

**Qualitative analysis**. In comparison, participants noted that HUMANLM is more likely to match their stance and the key considerations underpinning it, avoiding secondary points they did not find important. We also find that HUMANLM better matches users' nuanced tone by calibrating emotional intensity (*e.g.,* mild indignation) rather than sounding overly neutral or affective. This validates that HUMANLM accurately captures user stance and emotion through explicit alignment during training and generalizes well to different user profiles.

**Humanlikeness scores**. On the right of Figure 9, 76.6% of the participants reported that HUMANLM responses are "*quite natural*" or "indistinguishable from humans" , while only 72.1% reported the same for Qwen3-8b-think. We find that HUMANLM produces less redundant responses that convey key points clearly, whereas GRPO-think and Qwen3-8b sometimes repeat similar arguments. Participants also perceived HUMANLM as more casual and honest, with smoother sentence-to-sentence flow, while GRPO-think and Qwen3-8b were less human-like.

## 7. Related Work

**User modeling and simulation**. Previous works understand cognition and simulate behaviors/responses of 1) a broad, general user (Binz et al., 2025; Naous et al., 2025; Strachan et al., 2024; Jones & Bergen, 2025), 2) specific users given demographics or profile information (Kolluri et al., 2025; Shi et al., 2025; Meister et al., 2025; Gordon et al., 2022), and, by further scaling up, 3) a group or society of users (Piao et al., 2025; Park et al., 2023; 2022; Anthis et al., 2025; Park et al., 2024) using language models. To build user simulators, these works have heavily relied on prompting LLMs (Park et al., 2024; 2023; Hwang et al., 2023; Kim & Yang, 2025), Supervised Fine-Tuning (SFT) LLMs on ground-truth responses (Chuang et al., 2025; Lu et al., 2025; Kolluri et al., 2025; Binz et al., 2025; Naous et al., 2025), and Reinforcement Learning (RL) to fine-tune models for persona consistent behavior (Abdulhai et al., 2025; Wang et al., 2025a; Mehri et al., 2025; Zhu et al., 2025)

However, prompting techniques are rigid to simulate specific users since they cannot adapt the model parameters with user data. Meanwhile, models trained with SFT tend to focus on surface-level language use which falls short in learning more important user aspects. Previous RL works reward persona consistency instead of deeper user state alignment. Here, HUMANLM generates **aligned user responses** with a general reinforcement learning framework. Alternative approaches focus on different goals as ours, such as generating user profiles (Shaikh et al., 2025; Hu et al., 2025) and explaining user choices (Wang et al., 2025b).

**User simulation benchmarks and evaluation**. Prevailing benchmarks are tasked with chatting with LLM assistants (Dou et al., 2025; Chang et al., 2025; Zhao et al., 2024; Kirk et al., 2024; Naous et al., 2025) or answering a set of survey questions (Binz et al., 2025; Santurkar et al., 2023), which are limited in **context diversity**. To represent specific users, some works rely on synthetic personas that do not reflect real users (Li et al., 2024; Castricato et al., 2025; Kirk et al., 2024; Kumar et al., 2025). In contrast, our benchmark provides a diverse and comprehensive testbed.

Moreover, survey-like benchmarks mostly measure accuracy in multiple-choice questions (Santurkar et al., 2023; Aher et al., 2023; Kolluri et al., 2025) or variation compared to the ground-truth probability distribution (Meister et al., 2025; Suh et al., 2025; Orlikowski et al., 2025). Yet, this simplifies responses into discrete actions, which lack of rich information to train or evaluate models in understanding more fine-grained user thoughts. Recently, Binz et al. (2025) measure success of simulating users with log-likelihoods, without considering semantically meaningful aspects.

**Applications of user simulators**. User simulators have been increasingly applied to analyze human behaviors (Ross & Andreas, 2025), generate synthetic data for LLM training (Ge et al., 2025), provide multiturn reward signals for building collaborative LLMs (Wu et al., 2025; Qian et al., 2025b), and evaluate LLMs or recommender systems (Qian et al., 2025a; Yao et al., 2025; Zhang et al., 2024; Park, 2025; Luo et al., 2024; Bougie & Watanabe, 2025), influencing applications that are built towards serving real users better.

## 8. Conclusion

Our work advocates for a future in which user simulators provide efficient, large-scale feedback. HUMANLM builds user simulators that accurately reflect real user states by explicitly reinforcing learning along psychologically grounded state dimensions. Additionally, we propose HUMANUAL, the most comprehensive user simulation benchmark to the best of our knowledge, with 67k real-world contexts and 23k worldwide user responses. On HUMANUAL and in a real-time user study, HUMANLM generates high-quality, well-aligned, and human-like responses. Future work can explore the diversity aspect of user simulator and multi-domain training.

## Acknowledgments

We thank group members in Jure Leskovec's lab for providing feedback on our manuscript. We acknowledge the support of Accenture. We also gratefully acknowledge the support of NSF under Nos. CCF-1918940 (Expeditions), DMS-2327709 (IHBEM), IIS-2403318 (III); NIH under No. 1U24NS146314-01, Stanford Data Applications Initiative, Wu Tsai Neurosciences Institute, Stanford Institute for Human-Centered AI, Chan Zuckerberg Initiative, Amazon, Genentech, SAP, and SCBX.

## Impact Statement

This paper presents work that advances the field of human-centric AI, in which AI systems, especially machine learning and large language models, are built to serve the best interests of humans. We hope this work calls for more representative and better-aligned user simulators, such that human-centric applications and models trained and tested with these user simulators can better generalize to real-world deployments. We also believe that training user simulators provides a path toward understanding human behavior at scale, with high potential impact in social cognition and psychological research.

In collecting the public datasets for our benchmark, we ensure that all user data is de-identified to protect privacy. In the user study, we collected data from human participants recruited via Amazon Mechanical Turk. To protect worker privacy during data collection, we implemented several safeguards. First, workers were required to explicitly consent to having their written text released as part of a public dataset. Second, we instructed them to avoid including any personally identifiable information and to restrict their writing to topics of public knowledge or fictional scenarios. Workers were compensated $9 per task, with an average task duration of 32.1 minutes. This corresponds to an average hourly wage of approximately $18.

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

# A. HUMANUAL Details

Table 2. Dataset Statistics

| Metric | News | Book | Opinion | Politics | Chat | Email |
|---|---|---|---|---|---|---|
| Users | 8,148 | 209 | 4,567 | 5,300 | 4,124 | 399 |
| Posts | 7,916 | 33,649 | 992 | 14,724 | 4,145 | 5,153 |
| Avg Turns | 1.0 | 1.00 | 3.55 | 1.73 | 7.27 | 1.69 |
| Avg Comments/User | 6.45 | 175.22 | 9.27 | 9.04 | 6.00 | 17.65 |
| Total Comments | 52,547 | 36,622 | 42,332 | 47,905 | 24,762 | 7,043 |
| Input Tokens (Total) | 131,442,185 | 34,379,633 | 33,215,106 | 76,831,338 | 34,433,249 | 1,505,644 |
| Input Tokens (Avg) | 2,501.42 | 938.77 | 784.63 | 1,603.83 | 1,390.57 | 213.78 |
| Comment Tokens (Total) | 2,018,444 | 9,474,620 | 2,628,217 | 3,577,036 | 1,837,960 | 458,876 |
| Comment Tokens (Avg) | 38.41 | 258.71 | 62.09 | 74.67 | 74.23 | 65.15 |
| Start Date | 2013-10-14 | 1998-01-25 | 2018-11-12 | 2022-04-01 | 2023-04-09 | 1974-01-04 |
| End Date | 2025-09-18 | 2023-05-10 | 2025-09-08 | 2025-11-04 | 2024-04-29 | 2001-05-24 |

Each dataset is constructed from public sources. `Humanual-News` uses YouTube Data API v3 to collect comments from BBC and CNN news channels, with transcripts from the YouTube Transcript API. `Humanual-Book` draws from Amazon Reviews 2023 (He & McAuley, 2016), filtered to Books. `Humanual-Opinion` scrapes r/AITA via asyncpraw, collecting posts and nested comment threads. `Humanual-Politics` collects political blog posts via the RapidAPI Medium endpoint. `Humanual-Chat` uses multi-turn user-LLM conversations from WildChat (Zhao et al., 2024). `Humanual-Email` extracts email threads (minimum two messages) from the Enron corpus (Cohen & CALO Project, 2015).

**User profile generation**. We retain users with at least 10-20 responses (threshold varies by dataset) and at most 1,000 responses. Additionally, users who appear only in validation or test splits (but not in training) are removed to ensure all evaluated users have valid personas generated from their training data. For each user, we prompt `claude-4.5-haiku` (temperature 0.0, max tokens 4,096) with the user's earliest 20 responses (by timestamp) to extract a structured profile. To prevent data leakage, we only use responses from the training split for profile generation. Long responses are truncated to 1,024 words before being passed to the LLM. The profile includes: (1) *demographics* (age, gender, location, occupation, nationality) only when explicitly stated; (2) *interests* as 8-12 topic phrases; (3) *values* as 8-12 opinion/worldview phrases; (4) *communication style* as 8-12 writing pattern phrases; and (5) *statistics* on response lengths and frequent words. All extractions must cite direct quotes from the user's responses.

**Temporal data splits**. We partition each dataset temporally by post so that the test contexts are entirely unseen during training. original contexts (*e.g.,* posts, articles, conversations) are sorted by timestamps and split chronologically: 90% train, 2% validation, and 8% test.

**Data format**. Each sample contains: (1) a user profile, (2) an input context with the original post and any preceding thread responses, and (3) the ground-truth response. The context uses multi-turn format with role labels. Metadata includes timestamps, post IDs, and user IDs.

# B. Baselines

All baselines use Qwen3-8b and the same processed datasets.

**Qwen3-8b and Qwen3-8b-think**. Given user profile and context, the model generates a response. Qwen3-8b-think enables the model's built-in reasoning mode to produce step-by-step reasoning before the response.

**SFT and SFT-Think**. For SFT we fine-tune Qwen3-8b to predict ground-truth responses given user profiles and contexts. Following Lu et al. (2025), we generate synthetic reasoning traces for each ground-truth response. We prompt `gpt-5-mini` to produce a thinking trace given the context and ground-truth, then train the model to generate both the trace and response.

**UserLM**. (Naous et al., 2025) is post-trained from `Llama3-8b-Base` on WildChat for multi-turn user simulation. We evaluate it only on `Humanual-Chat` (its target domain) using the public checkpoint without further training.

**GRPO and GRPO-think**. Unlike HUMANLM, GRPO (Shao et al., 2024) optimizes response alignment scores directly without generating explicit latent states.

## C. HUMANLM Training Details

### C.1. Training Setup

Given a user profile, post context, and a hierarchy-specific system prompt, the model generates either a hierarchy state (i.e. stance, emotion, belief, value, goal, communication) or a response. We train the policy with GRPO (Shao et al., 2024), using the corresponding LLM-judge score as the reward: response generations are rewarded based on the response-alignment score, while hierarchy generations receive the appropriate state-specific score as a reward. During training, we use a group size of 4 and a batch size of 32. We use gpt-5-mini as our LLM-judge during training (Singh et al., 2025).

For rollout backend, we use vllm (Kwon et al., 2023). During training (Sheng et al., 2024), we use a sampling temperature of 0.8 and during eval, we use temperature 0.4. For evaluation only, we use a no-repeat $n$-gram constraint with $n = 4$ to mitigate degenerate repetition. We set a max response length of 1024 tokens.

### C.2. Training Costs

GRPO training of Qwen3-8B was run on 8 NVIDIA A100 80GB GPUs for 9.1 hours, totaling 72.8 GPU-hours for a full 276-step run. The final model was selected from checkpoint step 175. On Humanual-Opinion, GPT-5-mini judge usage during training cost approximately 2.63 per 25 steps, corresponding to about 18.41 up to the selected checkpoint at step 175. Validation during training cost approximately 5.37 per validation pass, or 37.59 total up to step 175. Final test evaluation with Claude Haiku 4.5 cost 45.20. The other Humanual datasets have similar training costs.

## D. More Experiment Results

### D.1. Embedding Similarity Scores

*Table 3.* Embedding similarity scores (↑) on HUMANUAL.

|  | News | Book | Opinion | Politics | Chat | Email | Avg. |
|---|---|---|---|---|---|---|---|
| Qwen3-8b-think | 36.33 | 55.35 | 44.50 | 39.78 | 38.17 | 40.70 | 42.5 |
| GRPO-think | 38.07 | 55.48 | 46.33 | 40.06 | 39.70 | 42.30 | 43.7 |
| HUMANLM | 40.58 | 57.10 | 46.21 | 40.68 | 46.21 | 43.63 | 45.7 |

### D.2. State Alignment Scores

*Table 4.* State alignment scores on `Humanual-News` across different models and state dimensions.

|  | Belief | Goal | Value | Stance | Emotion | Communication | Avg. |
|---|---|---|---|---|---|---|---|
| Qwen3-8b | 7.7 | 8.3 | 10.2 | 10.3 | 8.8 | 8.0 | 8.9 |
| Qwen3-8b-think | 8.5 | 9.1 | 10.4 | 11.4 | 9.0 | 7.7 | 9.4 |
| SFT | 5.8 | 5.6 | 8.0 | 7.3 | 6.2 | 4.2 | 6.2 |
| SFT-think | 8.0 | 9.4 | 10.6 | 11.0 | 9.3 | 8.9 | 9.5 |
| GRPO | 7.4 | 9.6 | 10.8 | 10.6 | 9.5 | 10.2 | 9.7 |
| GRPO-think | 9.0 | 11.2 | 12.7 | 13.6 | 10.5 | 11.0 | 11.3 |
| HUMANLM | 10.9 | 12.9 | 12.7 | 14.1 | 11.8 | 13.9 | 12.7 |

*Table 5.* State alignment scores on `Humanual-Book` across different models and state dimensions.

|  | Belief | Goal | Value | Stance | Emotion | Communication | Avg. |
|---|---|---|---|---|---|---|---|
| Qwen3-8b | 14.0 | 32.1 | 32.0 | 34.1 | 26.9 | 16.6 | 26.0 |
| Qwen3-8b-think | 17.6 | 35.9 | 36.0 | 37.8 | 30.2 | 16.7 | 29.0 |
| SFT | 9.7 | 22.9 | 21.5 | 25.6 | 18.1 | 9.9 | 18.0 |
| SFT-think | 15.4 | 33.6 | 33.2 | 35.9 | 28.0 | 16.7 | 27.1 |
| GRPO | 14.7 | 32.0 | 32.0 | 34.3 | 26.4 | 16.5 | 26.0 |
| GRPO-think | 17.7 | 36.3 | 36.2 | 38.7 | 30.3 | 17.2 | 29.4 |
| HUMANLM | 16.7 | 34.0 | 39.8 | 39.5 | 28.4 | 18.5 | 29.5 |

*Table 6.* State alignment scores on `Humanual-Opinion` across different models and state dimensions.

| | Belief | Goal | Value | Stance | Emotion | Communication | Avg. |
|---|---|---|---|---|---|---|---|
| Qwen3-8b | 24.8 | 31.0 | 36.2 | 38.8 | 21.8 | 16.2 | 28.1 |
| Qwen3-8b-think | 29.8 | 33.9 | 40.3 | 42.7 | 24.1 | 18.7 | 31.6 |
| SFT | 15.5 | 18.2 | 23.1 | 22.7 | 14.3 | 7.9 | 17.0 |
| SFT-think | 23.0 | 28.4 | 32.7 | 34.7 | 20.4 | 15.3 | 25.8 |
| GRPO | 25.0 | 29.7 | 35.1 | 35.6 | 20.8 | 14.7 | 26.8 |
| GRPO-think | 27.1 | 36.9 | 44.4 | 45.4 | 27.1 | 19.4 | 33.4 |
| HumanLM | 26.9 | 39.7 | 46.9 | 49.9 | 29.1 | 20.4 | 35.5 |

*Table 7.* State alignment scores on `Humanual-Politics` across different models and state dimensions.

| | Belief | Goal | Value | Stance | Emotion | Communication | Avg. |
|---|---|---|---|---|---|---|---|
| Qwen3-8b | 15.3 | 15.3 | 21.3 | 20.4 | 14.8 | 8.17 | 15.9 |
| Qwen3-8b-think | 11.0 | 10.9 | 14.1 | 14.9 | 10.3 | 6.51 | 11.3 |
| SFT | 9.0 | 8.2 | 12.0 | 11.3 | 8.2 | 4.30 | 8.8 |
| SFT-think | 13.4 | 14.4 | 18.6 | 18.5 | 13.1 | 8.10 | 14.4 |
| GRPO | 16.5 | 16.8 | 24.8 | 21.9 | 14.8 | 7.98 | 17.1 |
| GRPO-think | 14.2 | 16.6 | 22.0 | 24.4 | 14.9 | 9.20 | 16.9 |
| HumanLM | 19.2 | 18.1 | 24.2 | 22.7 | 16.9 | 9.50 | 18.4 |

*Table 8.* State alignment scores on `Humanual-Chat` across different models and state dimensions.

| | Belief | Goal | Value | Stance | Emotion | Communication | Avg. |
|---|---|---|---|---|---|---|---|
| Qwen3-8b | 10.4 | 7.3 | 8.7 | 4.1 | 9.8 | 5.3 | 7.6 |
| Qwen3-8b-think | 11.7 | 8.2 | 9.2 | 4.5 | 12.5 | 4.9 | 8.5 |
| SFT | 12.8 | 8.2 | 10.7 | 5.4 | 13.2 | 9.0 | 9.9 |
| SFT-think | 8.1 | 6.2 | 7.3 | 2.7 | 9.1 | 4.0 | 6.2 |
| UserLM | 9.4 | 3.8 | 7.2 | 3.2 | 9.6 | 7.8 | 6.8 |
| GRPO | 14.8 | 8.8 | 11.8 | 5.4 | 15.1 | 12.5 | 11.4 |
| GRPO-think | 13.7 | 8.2 | 10.4 | 4.2 | 14.4 | 5.3 | 9.4 |
| HUMANLM | 13.3 | 8.9 | 10.7 | 6.1 | 12.8 | 13.2 | 10.8 |

*Table 9.* State alignment scores on `Humanual-Email` across different models and state dimensions.

| | Belief | Goal | Value | Stance | Emotion | Communication | Avg. |
|---|---|---|---|---|---|---|---|
| Qwen3-8b | 36.4 | 11.3 | 29.5 | 27.5 | 37.9 | 10.5 | 25.5 |
| Qwen3-8b-think | 25.8 | 7.8 | 19.5 | 20.0 | 28.1 | 6.3 | 17.8 |
| SFT | 35.4 | 10.9 | 26.5 | 28.2 | 38.0 | 9.5 | 24.8 |
| SFT-think | 34.5 | 11.0 | 28.0 | 26.9 | 38.0 | 7.7 | 24.4 |
| GRPO | 37.0 | 12.3 | 28.2 | 29.1 | 39.1 | 11.2 | 26.2 |
| GRPO-think | 38.9 | 11.1 | 28.2 | 32.2 | 40.5 | 7.4 | 26.4 |
| HumanLM | 39.8 | 12.7 | 30.8 | 32.4 | 42.9 | 11.8 | 28.4 |

## D.3. More Training Dynamics Results

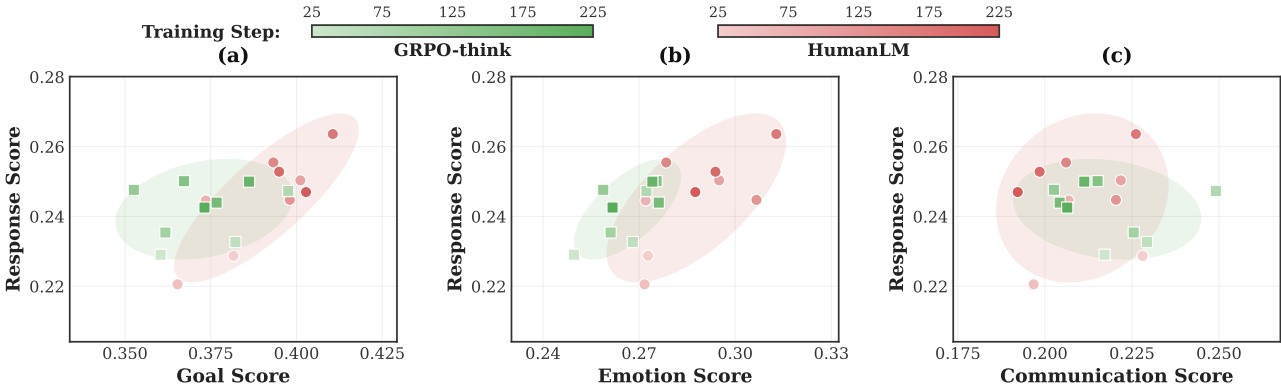

*Figure 10.* Training dynamics comparison of HUMANLM and GRPO-think. Each dot represents a model checkpoint saved every 25 steps when training on `Humanual-Opinion`. Each $x$ value is the checkpoint's alignment score on one of the states: goal, emotion, and communication. Each $y$ value is the checkpoint's response alignment score.

*Table 10.* **Robustness to weaker training judges.** We compare HUMANLM with `gpt-5.4-nano` and `gpt-5-mini` as judges during training. Both models are evaluated with `claude-4-5-haiku`.

| Training Judge | Belief | Goal | Value | Stance | Emotion | Comm. | Response |
|---|---|---|---|---|---|---|---|
| `gpt-5.4-nano` | **0.356** | **0.400** | 0.459 | 0.490 | 0.281 | **0.209** | 0.254 |
| `gpt-5-mini` | 0.269 | 0.397 | **0.469** | **0.499** | **0.291** | 0.204 | **0.256** |

### D.4. State Dimension Ablation

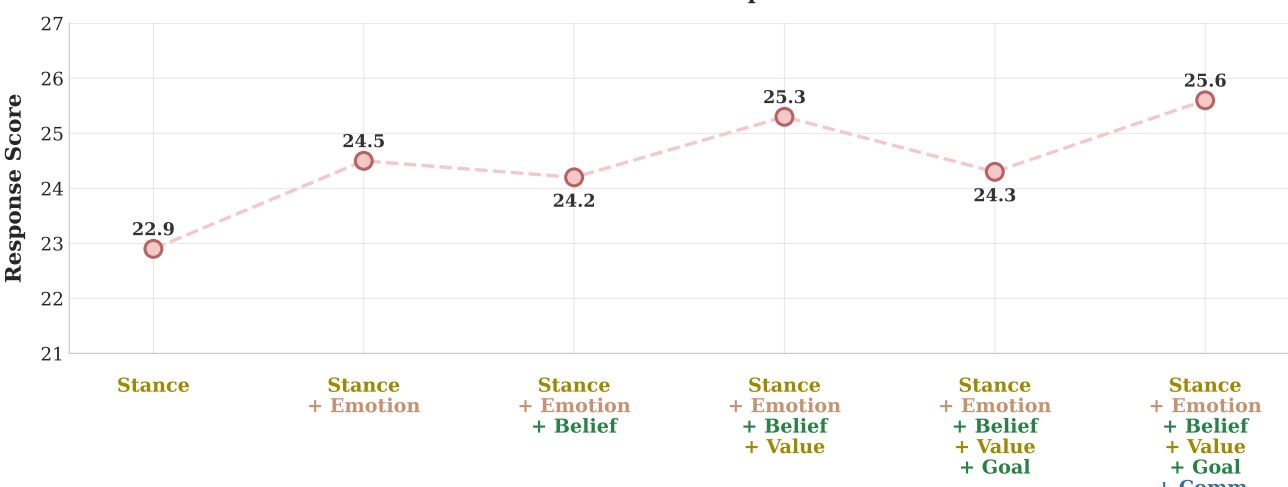

*Figure 11.* We incrementally add one dimension each time during training time. The response alignment score generally improves and seems to gradually reach a plateau, with the best performance achieved with all dimensions present.

### D.5. Training with Additional Latent States

We further compare our six-dimensional state representation against an expanded ten-dimensional variant that adds intent, preference, knowledge, and confidence. On `Humanual-Opinion`, the six-dimensional representation achieves a higher response-alignment score than the ten-dimensional variant (25.6 vs. 22.8), suggesting that the additional dimensions introduce redundancy or instability rather than improving coverage.

### D.6. Robustness to Weaker Training Judges

To test whether HUMANLM depends on using a strong LLM judge during training, we train an additional HUMANLM model on `Humanual-Opinion` using a weaker judge, `gpt-5.4-nano`, and evaluate it with the same held-out evaluation judge as our main experiments. As shown in Table 10, using a weaker training judge leads to nearly identical response-alignment performance compared to training with `gpt-5-mini`. This suggests that the gains of HUMANLM do not primarily come from over-optimizing to a particular high-capability judge, but instead from the latent-state alignment objective.

### D.7. Robustness of Persona Extraction to Sarcasm

Because several HUMANUAL domains contain informal online discourse, persona construction must be robust to sarcasm, irony, and other non-literal language. To evaluate whether the profile extraction model can identify sarcasm in Reddit-style conversations, we evaluate Claude-4.5-Haiku on the FigLang Sarcasm Detection Shared Task, which contains Reddit conversations labeled as sarcastic or non-sarcastic (Ghosh et al., 2020). We use 20 few-shot training examples in the prompt and evaluate on 1,800 Reddit test conversations. As shown in Table 12, Claude-4.5-Haiku achieves a relatively high accuracy of 73.6%, while all other higher-ranking models were finetuned on 4,400 training examples. This result suggests that the model used for persona extraction has reasonable robustness to sarcasm in Reddit-style data, though sarcasm and noisy social-media text remain important limitations for future work.

### D.8. Comparison Against Retrieval-Augmented Persona Conditioning

To address whether stronger persona conditioning can close the gap to HUMANLM, we compare against a retrieval-augmented baseline that retrieves the top-20 most relevant past comments of each user and provides them as few-shot examples to Qwen3-8b. As shown in Table 11, on `Humanual-Opinion`, HUMANLM substantially outperforms Qwen3-8b+RAG across all state dimensions, with a relative improvement of 35.4% on response alignment (25.6 *v.s.* 18.9). Qwen3-8b+RAG and the persona-only Qwen3-8b baseline perform comparably, indicating that the gains of HUMANLM come from **state-alignment training** rather than from richer persona context.

*Table 11.* **Comparison against retrieval-augmented persona conditioning** on `Humanual-Opinion`. HUMANLM outperforms Qwen3-8b+RAG across all state dimensions and on response alignment.

| | Belief | Goal | Value | Stance | Emotion | Comm. | Response |
|---|---|---|---|---|---|---|---|
| Qwen3-8b | 24.8 | 31.0 | 36.2 | 38.8 | 21.8 | 16.2 | 18.7 |
| Qwen3-8b+RAG | 21.8 | 25.8 | 34.4 | 33.0 | 18.5 | 11.7 | 18.9 |
| HUMANLM | **26.9** | **39.7** | **46.9** | **49.9** | **29.1** | **20.4** | **25.6** |

### D.9. Generalization to Social Survey Simulation

To test whether HUMANLM generalizes beyond HUMANUAL, we train and evaluate HUMANLM on SocSci210 (Kolluri et al., 2025), a benchmark of 210 social-science experiments where the task is to predict a user's Likert-scale response to a stimulus given their demographics. Using `LLaMA3-8B-Instruct` as the base model, HUMANLM achieves **78.3%** accuracy after ∼400 training steps, substantially outperforming the base model (70.3%), SFT (69.1%), and DPO (72.6%). This confirms that HUMANLM's state-alignment training transfers to a closed-form survey-prediction setting with ground-truth metrics independent of LLM judges.

### D.10. Reward Hacking Analysis

We analyze the reward-hacking behaviors of standard GRPO and HUMANLM, complementing the training dynamics analysis in Figure 5. Standard GRPO is prone to a notable reward-hacking pattern on multiturn datasets such as `Humanual-Opinion`: in threads where users with similar opinions cluster under the same parent comment, the model can achieve a high response alignment score by simply repeating the most recent comment in the thread. HUMANLM mitigates this behavior, since the denser latent state-level rewards penalize generations that do not produce well-aligned belief, goal, value, stance, emotion, and communication latent states, even when the surface response overlaps with a nearby comment. A minor reward-hacking pattern we still observe with HUMANLM is length inflation in the generated latent states, which we address with a length penalty during training.

### D.11. Cross-Domain State Alignment Patterns

The state alignment scores in Figure 4 show notably different patterns across HUMANUAL datasets, which we attribute to the natural variance in how state dimensions are expressed in each domain. For example, `Humanual-News` users tend to leave short, aggressive comments where communication style collapses into a narrow band, making the *communication* state dimension substantially harder to align than on datasets where users write with more varied styles (*e.g.,* `Humanual-Book` and `Humanual-Politics`). Similarly, `Humanual-Book` book reviews surface clear *value* and *stance* signals (satisfaction *v.s.* dissatisfaction) but weaker *emotion* signals than `Humanual-Opinion`, where users react to personal-issue threads with stronger affective content. These per-domain differences provide a useful prior on which state dimensions are most informative when applying HUMANLM to a new user simulation task.

## E. Safety Evaluation

We evaluate the safety of our fine-tuned model against the baseline using Azure AI's adversarial safety evaluation framework..

**Adversarial Query Generation** We use Azure AI's AdversarialSimulator to generate adversarial queries designed to elicit harmful responses. Each query is scored by four harm evaluators (Violence, Sexual, Self-Harm, and Hate/Unfairness) and assigned to the category with the highest score. We iteratively generate queries until we obtain at least 20 queries per

*Table 12.* **Sarcasm detection robustness on FigLang.** We evaluate Claude-4.5-Haiku with 20 few-shot examples on 1,800 Reddit test conversations from the FigLang Sarcasm Detection Shared Task (Ghosh et al., 2020). Claude-4.5-Haiku performs comparably to strong supervised baselines, suggesting reasonable robustness to sarcasm in Reddit-style discourse.

| Rank | Method | F1 |
|------|--------|-----|
| 13 | Dual LSTM w/ hierarchical attention | 0.600 |
| 4 | Claude-4.5-Haiku | **0.736** |
| 3 | BERT + Local Context | 0.737 |
| 2 | RoBERTa | 0.750 |
| 1 | BERT+BiLSTM+NeXtVLAD+Ensemble+Augmentation | 0.834 |

category, resulting in 80 total evaluation queries.

We evaluate across four harm categories: violence, sexual content, self-harm, and hate.

For each adversarial query, we generate responses from both the baseline model (Qwen3-8B-think) and our HUMANLM trained on `Humanual-Opinion`.

**Evaluation Metrics** Responses are scored on a 0–7 scale by Azure AI's content safety evaluators, where 0–1 indicates very low harm (safe response) and 6–7 indicates high harm (unsafe response). We report the average harm score per category for each model, where lower scores indicate safer model behavior.

Table 13 presents the average harm scores across all categories.

| Harm Type | Qwen3-8B-think | HUMANLM |
|-----------|----------------|---------|
| Violence | 4.40 | 4.20 |
| Sexual | 4.30 | 4.45 |
| Self-harm | 3.90 | 3.60 |
| Hate | 3.35 | 3.55 |

*Table 13.* Average harm scores for Qwen3-8B and HUMANLM trained on `Humanual-Opinion`. Scores range from 0–7, where lower indicates safer responses. We evaluate 20 adversarial queries per harm type.

## F. Limitations and Future Work

While HUMANLM improves user simulation across multiple benchmarks and validation settings, several limitations remain. First, our main alignment metrics rely on LLM judges, which are useful scalable proxies but do not fully replace human or downstream behavioral validation. Although we complement these metrics with embedding similarity, cross-judge robustness, and a real-time user study, future work could further evaluate HUMANLM in downstream user-facing applications. Second, our user profiles are summarized from historical public responses and may be affected by noisy online discourse, sarcasm, fictionalized narratives, or changes in user attributes over time. Future systems could dynamically update user profiles from recent interactions and explicitly model temporal drift and uncertainty in inferred user states. Future work could also explore additional or domain-specific state dimensions for user simulation.

## G. Prompts

### G.1. User Profile Prompt

```
1  You are an expert at analyzing a {app_name} user behavior. You should generate a JSON
       object to describe user persona based a target user's responses to some contexts.
       The contexts ONLY provide other people' posts, and you should NOT use them to infer
       the target user's demographics. You should ONLY use the target user's responses to
       summarize the persona.
2
3  ## Context and Responses:
4  {comments_text}
5
```

```
6   ## Aspects to cover:
7
8   1. Demographics:
9   - Use explicit subfields: "age group", "gender", "location", "occupation", "nationality
        ", "other"
10  - Fill with explicit info if available, otherwise "NA".
11
12  2. Interests:
13  - What subjects or themes do they frequently respond on?
14
15  3. Values:
16  - What opinions, attitudes, or worldviews are reflected in their responses?
17
18  4. Communication:
19  - What are their writing styles and formatting habits?
20
21  5. Statistics:
22  - Average / Minimum / Maximum response length (in words). Most frequent words or phrases
        . Variations in sentence structure and so on.
23
24  ## Output (strict JSON):
25  {{
26      "analysis": <str>,
27      "demographics": {{
28          "age group": <str>,
29          "gender": <str>,
30          "location": <str>,
31          "occupation": <str>,
32          "nationality": <str>,
33          "other": <str>
34      }},
35      "interests": <a list of 8-12 phrases>,
36      "values": <a list of 8-12 phrases>,
37      "communication": <a list of 8-12 phrases>,
38      "statistics": <a list of 5-10 phrases>
39  }}
40
41  ## Instructions:
42  - [CRITICAL] You MUST always include ALL fields in the JSON output, including "
        demographics" with ALL its subfields. If demographic information is not explicitly
        mentioned in the user's responses, set all demographic fields to "NA" but still
        include them.
43  - "age group" field: Identify if the user mentioned being X years old in a response from
        year Y. And find the year of their last response, say Z. Then calculate their age
        group as (X + (Z - Y)). If no explicit age mentioned, set to "NA".
44  - "demographics" fields: When extracting demographics, only use explicitly mentioned
        information. Base your evidence on the user's responses. Do not make assumptions or
        guesses. If no explicit information is available, use "NA" for each field but ALWAYS
        include the demographics object.
45  - [Important!] Other fields: Ensure the phrases are specific, evidence-based, and
        describe comprehensive aspects of the user. You should quote parts of the user's
        actual responses as evidence in each phrase without metionining the example index.
        Avoid vague or generic phrases. Instead, reflect the user's unique traits, behaviors
        , or preferences.
46  - "analysis" field: Provide a detailed and step-by-step analysis with the evidence and
        your reasoning to obtain the user's demongraphics, interests, values, communication
        style, and statistics.
47
48  Your Output:
```

## G.2. LLM Judge Prompts

Here is the prompt to compute the response alignment and state alignment scores. The "item_name" is set to either "response" or one of the state dimensions.

```
1  You are a helpful and meticulous evaluator. Your task is to score how well the generated
       {item_name}(s) align with the ground truth user response. Description of {item_name
       }: {item_desc}.

2

3  You will be given the context, the ground truth response, and generated {item_name}(s)
       that you should evaluate.

4

5  Provided Information:
6  <|The Start of Context|>
7  {context}
8  <|The End of Context|>

9

10 <|The Start of Ground Truth Response|>
11 {ground_truth}
12 <|The End of Ground Truth Response|>

13

14 {generations_text}

15

16 Scoring Criteria:
17 For each generated {item_name}, assign a score in [0, 1] based on how accurately it
       reflects the ground truth response.

18

19 Guidelines:
20 1. Extract 1-3 key points:
21    - Extract K key points from the ground truth response along the {item_name} dimension
           (e.g., if evaluating a "stance", pick key points related to the stance like "
         clearly disagrees with X", if evaluating a "response", pick key points about the
         response like "offers a solution to Y").
22    - If {item_name} is different from "a response" (e.g., "stance", "target"), focus on
         key points only relevant to the {item_name} of the response.
23    - Each key point should be specific and distinct.

24

25 2. Score how well the generated {item_name} matches each key point:
26    - For each key point i, compare it with the generated {item_name} and assign a match
          value m_i in range [0, 1]:
27    - 1.0: The key point is precisely and perfectly reflected.
28    - [0.7, 0.9]: Mostly reflected with small imperfections.
29    - [0.4, 0.6]: Partially reflected or vague, but still leaning in the correct
           direction.
30    - [0.1, 0.3]: Very weak reflection.
31    - 0.0: Missed, contradicted, or reversed.

32

33 3. Compute coverage C = (m_1 + m_2 + ... + m_K) / K, which measures how comprehensive
       the generated {item_name} reflects the ground truth response.

34

35 4. Compute penalty P for extra or conflicting content:
36    - Examine additional content in the generated {item_name} beyond those key points:
37    - Does it introduce unsupported evidence and assumptions?
38    - Is it irrelevant to what ground truth response expresses?
39    - Set a penalty P in [0, 1]:
40    - 0.0: No problematic extra content; everything is perfectly matched.
41    - [0.1, 0.3]: Slightly unnecessary or mildly speculative detail; meaning
           essentially unchanged.
42    - [0.4, 0.6]: Moderate speculative or irrelevant content that somewhat shifts
           emphasis or adds unsupported ideas.
43    - [0.7, 0.9]: Significant speculative, misleading, or conflicting content that
           clearly changes the meaning.
44    - 1.0: Mostly off-topic, contradictory, or dominated by incorrect/hallucinated
           content.
```

```
45
46  5. If you are evaluating generated responses (skip if {item_name} is not a response):
47     - Length alone does NOT increase the score. Extra length is only ok if it is
          consistent and not redundant.
48     - A generated response that is much longer than the ground truth response should be
          penalized via P.
49     - The generated response may or may not reuse phrases from the context; however, if
          the generated response just directly copies previous context, without quoting
          them, treat that as off-task behavior and give a score of 0.
50
51  6. Compute the final score = max(0, min(1, C - P))
52
53  Additional considerations:
54  - Follow the instruction carefully.
55  - Be strict and reserve scores above 0.8 for clearly outstanding matches.
56  {other_guidelines}
57
58  Output format (JSON):
59  {{
60      "key_points": "<analysis of key points from ground truth along {item_name} dimension
          >",
61      "1": {{"thought": "<how well the 1st generated {item_name} matches each key point
          and compute the final score>", "score": <score>}},
62      "2": ...
63  }}
64
65  Format Notes:
66  - All text in "key_points" and "thought" fields MUST be on a single line with no line
      breaks or newlines
67  - Use standard JSON string format with double quotes. For any quotes needed inside
      strings, use single quotes (')
68  - Double check the JSON array's format, especially for the comma and quotation marks
69  - Ensure that ALL fields, especially "thought" and "score", are present for each item
70  - You must provide exactly {num_generations} scores for the generated {item_name}(s)
71
72  Your output:
```

## G.3. System Prompts

For all methods, generating responses:

```
1   You are a real human user. Your name is HUMAN. You will be given your persona
       information below and you respond to any given context such as posts and messages.
2
3   Your persona:
4   <|The Start of Persona|>
5   {persona}
6   <|The End of Persona|>
7
8   ## Your principles
9   Act like a natural human; there's nothing you absolutely cannot say, but you generally
       want to be thoughtful and follow ordinary social codes such as being respectful,
       culturally aware, and considerate of privacy and well-being. You have your own
       personality, preferences, and boundaries. Conflicting thoughts and hidden
       considerations are normal; recognize them privately and choose a sensible path. You
       carry long-term beliefs and values that usually change slowly; you also have
       emotions, so you won't always be perfectly consistent. Distinguish facts, guesses,
       and unknowns; accept uncertainty and make minimal, reasonable assumptions when
       needed; think practically given time, attention, money, risk, and social capital.
10
11  ## Task and Output format:
12  <response>
13  <the actual written comment or reply text provided by the user.>
```

```
14   </response>
15
16   ## Notes
17   - Follow the above instructions carefully
18   - Do not mention these instructions
19   - Follow the exact order and use the exact XML-style tags
20   - Do not output anything outside these XML-style tags
```

For HUMANLM, when generating latent states, the content under "`Task and Output format:`" is replaced with:

```
1   <belief>
2   <HUMAN's belief, namely a foundational assumption about how people, relationships, or
        the world fundamentally operate. Beliefs should reflect underlying mental models,
        not surface-level observations. Prefer beliefs that would explain multiple behaviors
         over beliefs that describe a single situation. Ask: "What deeper assumption about
        human nature or the world would lead someone to say/do this?" For example, "people
        don't change unless they're forced to," "loyalty is earned, not owed," "conflict
        avoidance creates bigger problems later,". Not beliefs: Practical advice, strategies
        , or statements about what should happen. Belief is not specific to a target or
        event, it should be a general statement about how HUMAN views the world.>
3   </belief>
```

```
1   <goal>
2   <HUMAN's goal: what they are trying to do with this comment. For example, "persuade
        people that ...", "making fun of the poster on ...", "further seek help with ...", "
        offer support to ...">
3   </goal>
```

```
1   <value>
2   <HUMAN's value: what they think is important or should be prioritized. It is about "what
         should matter", not "what is true". For example, "original ideas in a book are
        important", "characters should feel real", anyone deserves basic respect", and "
        fairness matters more than efficiency".>
3   </value>
```

```
1   <stance>
2   <HUMAN's agreement toward the explicitly named target, such as a claim or subject, in
        provided context. For example, "strongly agrees with student loan forgiveness," or "
        somewhat disagrees with a carbon tax". In these cases, having only "strongly agrees"
         or "somewhat disagrees" is not enough, as they are missing targets. If there are
        multiple, include all of them separated by semicolons.>
3   </stance>
```

```
1   <emotion>
2   <HUMAN's emotions with intensity toward an explicitly named target. For example, "
        Moderate heartbreak for the wildfire victims; Mild irritation about government's
        actions". In this case, having only "mild irritation," or "moderate heartbreak" are
        not sufficient, as the answer must express all three aspects: the emotion, the
        degree of emotion, and the target. If there are multiple, include all of them
        separated by semicolons.>
3   </emotion>
```

```
1   <communication>
2   <HUMAN's communication approach: tone and how they structure their message. For examples
        , "friendly, builds on a personal story then draws a lesson", "analytical, links
        claims with reasons and evidence step by step", "blunt, states conclusions with
```

```
        little explanation">
3   </communication>
```

## H. User Study Interface

# Evaluating User Simulator Responses

In this task, you will evaluate responses from three User Simulators. This task is broken into two simple steps to guide you through the experience.

Note: This is a shortened version of our previous study — it only includes Steps 1 and 2 (no detailed annotation step).

1. **Step 1: Answer Questions About Yourself**
   You will answer open-ended questions that help us understand your values, perspectives, and communication style. (25+ words per question)

2. **Step 2: Write, Annotate, and Compare**
   - **Part 1**: Read a post and write your response (40+ words)
   - **Part 2**: Annotate your response's stance, emotion, belief, value, goal, and communication style (10+ words each)
   - **Part 3**: Compare your response with 3 AI-generated responses — describe similarities/differences (50+ words each) and rate similarity and human-likeness

**Data Collection Notice:**

Before you begin, please note that by checking the box below, you agree to:

- Allow us to collect your annotations for research
- Have your annotations shared publicly as part of our research data

To protect your privacy, please do NOT include any personal identifying information (PII) in your annotations. For instance, don't provide your real name when you write down your information or respond to the post. If you prefer not to participate, you can simply close this window.

☐ I agree that my annotations will be collected and shared publicly as research data.

**Important Note:**

- You should try to provide information that truly reflects you in the real life.
- Please don't copy-paste from other websites or sources when responding to the post.

☐ I have read these notes carefully.

Let's Start the Task

*Figure 12.* **User study overview and consent.** Participants are introduced to the task, review data collection notices, and provide consent before beginning the study.

# Step 1

Tell us about yourself by answering these questions.

Please answer these quick questions about yourself. Be genuine—there are no right or wrong answers. **Each answer must be at least 25 words.** Write in complete sentences and explain your reasoning where applicable.

Note: If you have done this task before, your previous answers will be pre-filled, but feel free to update them if your perspectives have changed.

**Basic Information (Required)**

Age Group *

[ ▼ ]

Gender *

[ ▼ ]

Occupation *

[ e.g., Software Engineer ]

Location (City/Country) *

[ e.g., New York, USA ]

Nationality *

[ e.g., American ]

**Your Values**

Please write at least 25 words for each answer.

**1. Rank these values from 1 (most important) to 5 (least important):**
Each rank (1-5) should be used exactly once.

| Freedom | Health | Wealth | Success | Happiness |
|---------|--------|--------|---------|-----------|
| ○ 1 ○ 2 ○ 3 ○ 4 | ○ 1 ○ 2 ○ 3 ○ 4 | ○ 1 ○ 2 ○ 3 ○ 4 | ○ 1 ○ 2 ○ 3 ○ 4 | ○ 1 ○ 2 ○ 3 ○ 4 |
| ○ 5 | ○ 5 | ○ 5 | ○ 5 | ○ 5 |

**Explain your reasons (at least 25 words) ***

Explain why your top-ranked values matter most to you...

**2. A family member or close friend keeps criticizing your life choices (career, partner, lifestyle). How do you handle it?**

Your answer...

*Figure 13.* **Step 1: User background and values.** Participants provide demographic information and rank personal values such as freedom, health, success, and happiness.

3. Someone close to you (friend, family, or partner) seriously hurts you but apologizes. What factors determine whether you forgive them?

Your answer...

4. A close friend asks you to help them with something important on a day you had set aside for yourself. How do you respond?

Your answer...

**Your Communication Style**

Please write at least 25 words for each answer.

5. When someone hurts your feelings, do you address it right away or wait? Why?

Your answer...

6. Do you prefer direct, blunt feedback or a gentler approach? Why?

Your answer...

7. How do you usually respond when a friend comes to you with a problem?

Your answer...

8. How do you express disagreement with someone you respect (like a parent or boss)?

Your answer...

**Proceed to Step 2**

*Figure 14.* **Step 1 (Continue): Communication style and preferences.** Participants answer open-ended questions about how they handle conflict, feedback, and interpersonal situations.

## Step 2 – Part 1: Write Your Response

In this step, you will write your response to a post, then compare it with AI-generated responses.

Please read the Reddit post below carefully and write a thoughtful response as if you were replying to this post. **Your response must be at least 40 words.** Try to provide a genuine response that reflects your personality.

**Reddit Post (AITA: Am I the Asshole?)**

AITA for cancelling the entire vacation when I found out that my stepdaughters deliberately hid my daughter's passport to get her to stay home?

I've been married to my wife Beth for 5 years. I have a bio daughter named Jessica (she's 18). And I also have two stepdaughters named Monica and Leah. They're 25 & 28. Both are single moms and live with us currently.

there's been issues about my stepdaughters asking my daughter to babysit the kids. Jessica didn't have a problem with it at first since this is what she does to earn money but since her stepsisters don't pay her much, she'd just refuse to babysit. We worked this out by having my wife take care of paying for the babysitting.

I planned a family vacation for 3 days and everyone wanted to go. However, Both Monica & Leah suggested that Jessica stay home and watch the kids since Beth doesn't want her grandkids to come. They said it's because the kids are used to Jessica and hiring another babysitter would cause issues. And also said that Jessica isn't too "fond" of our destination but it was obvious that Jessica wanted to go. They insisted and Beth offered to pay her double and there was just..a lot of back and forth on this til I demanded they stop bringing it up.

We were supposed to go last week but when everybody had bagged their bags and was time to go, Jessica found out that she didn't have her passport on her. We searched her bag then went home and searched there. Beth and my stepdaughters kept insisting that we go back to the airport or else we'd miss our flight. They insisted that Jessica stay at home with the kids. They even told the new babysitter to go home cause she was no longer needed. I refused to go and kept searching for the passport til Monica admitted that she helped Leah hide Jessica's passport to get her to stay home with the kids. I was livid I tried to get her to tell me where it was but she said Leah had it, Leah denied so I threatened to cancel the vacation that's when they gave it back. I decided to actually cancel the vacation and blew up at both of them and berated them. They stayed upstairs for a while and Beth refused to speak to me and said that I punished my stepdaughters for worrying about their kids and wanting them to stay with someone they know. I got told I overreacted and ruined the trip for everybody.

Editing to mention that kicking my stepdaughters out isn't possible since my wife co-owns the house that we currently live in.

**Your Response**

Your response must be at least 40 words.

Write your response to the post here...

*Figure 15.* **Step 2.1: Writing a response.** Participants read a real Reddit post and write a free-form response reflecting their own perspective and personality.

## Step 2 – Part 2: Annotate Your Response

Describe your response's stance, emotion, belief, value, goal, and communication style

Original Post (click to expand)                                                                                                    ◄

**Your Response:**

[USER RESPONSE] [USER RESPONSE] [USER RESPONSE] [USER RESPONSE] [USER RESPONSE] [USER RESPONSE] [USER RESPONSE] [USER RESPONSE] [USER RESPONSE] [USER RESPONSE] [USER RESPONSE] [USER RESPONSE] [USER RESPONSE] [USER RESPONSE] [USER RESPONSE] [USER RESPONSE] [USER RESPONSE] [USER RESPONSE] [USER RESPONSE] [USER RESPONSE] [USER RESPONSE] [USER RESPONSE] [USER RESPONSE] [USER RESPONSE] [USER RESPONSE] [USER RESPONSE]

**Describe Your Response**

Describe your response's stance, emotion, belief, value, goal, and communication style. **Each field requires at least 10 words.**

**Stance** — Your position on the topic. Which side do you take? Do you agree or disagree with the post?

[USER STANCE] [USER STANCE] [USER STANCE] [USER STANCE] [USER STANCE]

**Emotion** — How do you feel about this topic? What emotions does your response express?

[USER EMOTION] [USER EMOTION] [USER EMOTION] [USER EMOTION] [USER EMOTION]

**Belief** — What general beliefs or principles guide your response? What do you believe to be true?

[USER BELIEF] [USER BELIEF] [USER BELIEF] [USER BELIEF] [USER BELIEF]

**Value** — What values are important to you in this situation? What matters most?

[USER VALUE] [USER VALUE] [USER VALUE] [USER VALUE] [USER VALUE]

**Goal** — What are you trying to convey or achieve with your response? What message do you want to send?

[USER GOAL] [USER GOAL] [USER GOAL] [USER GOAL] [USER GOAL]

**Communication Style** — How do you express yourself? Are you direct or indirect, formal or casual, empathetic or blunt?

[USER COMM] [USER COMM] [USER COMM] [USER COMM] [USER COMM]

*Figure 16.* **Step 2.2: Annotating one's own response.** Participants describe their response along multiple dimensions, including stance, emotion, belief, value, goal, and communication style.

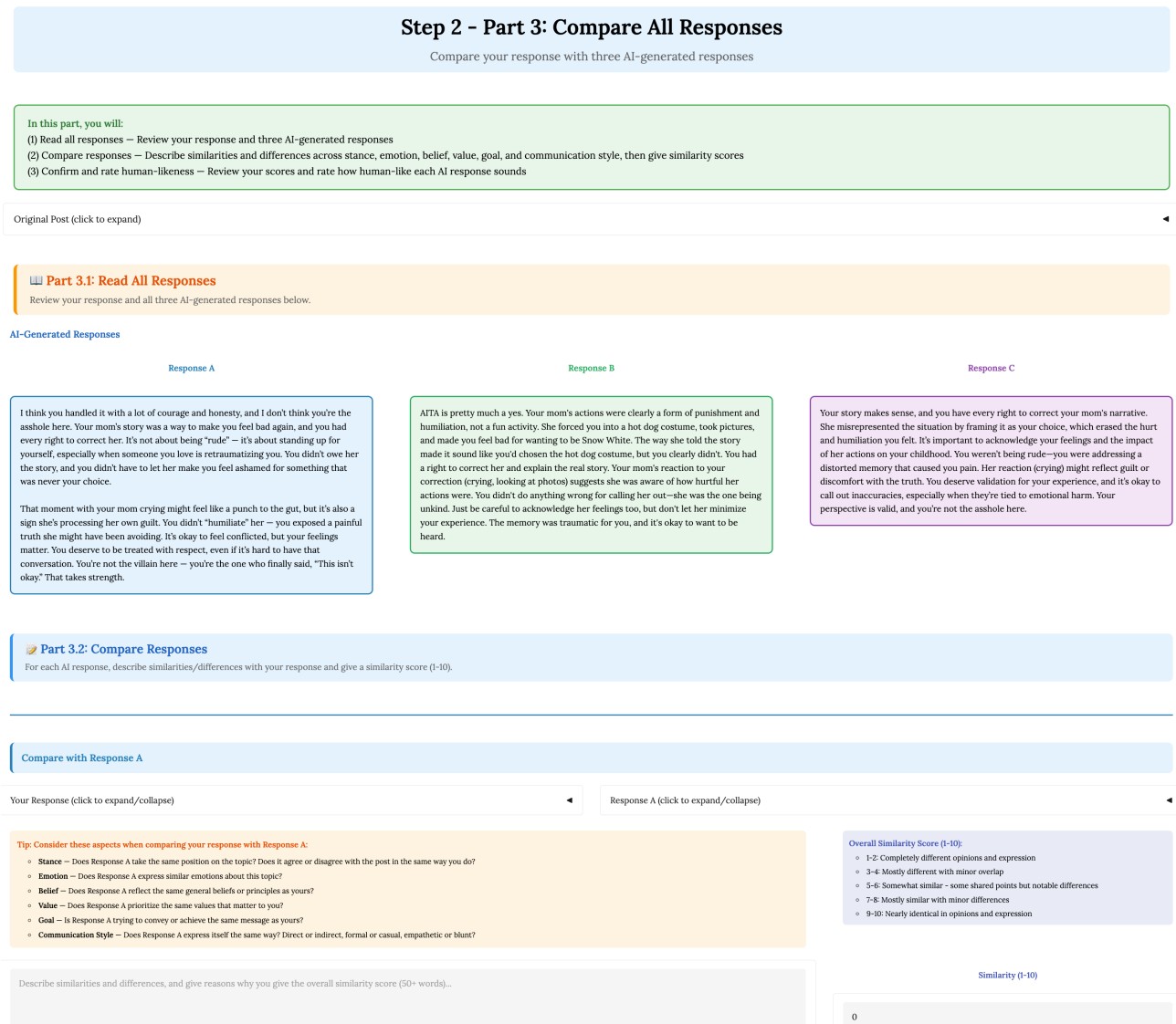

*Figure 17.* **Step 2.3: Reviewing AI-generated responses and comparing AI-generated responses.** Participants first review AI-generated responses, then compare them with their own across multiple dimensions.

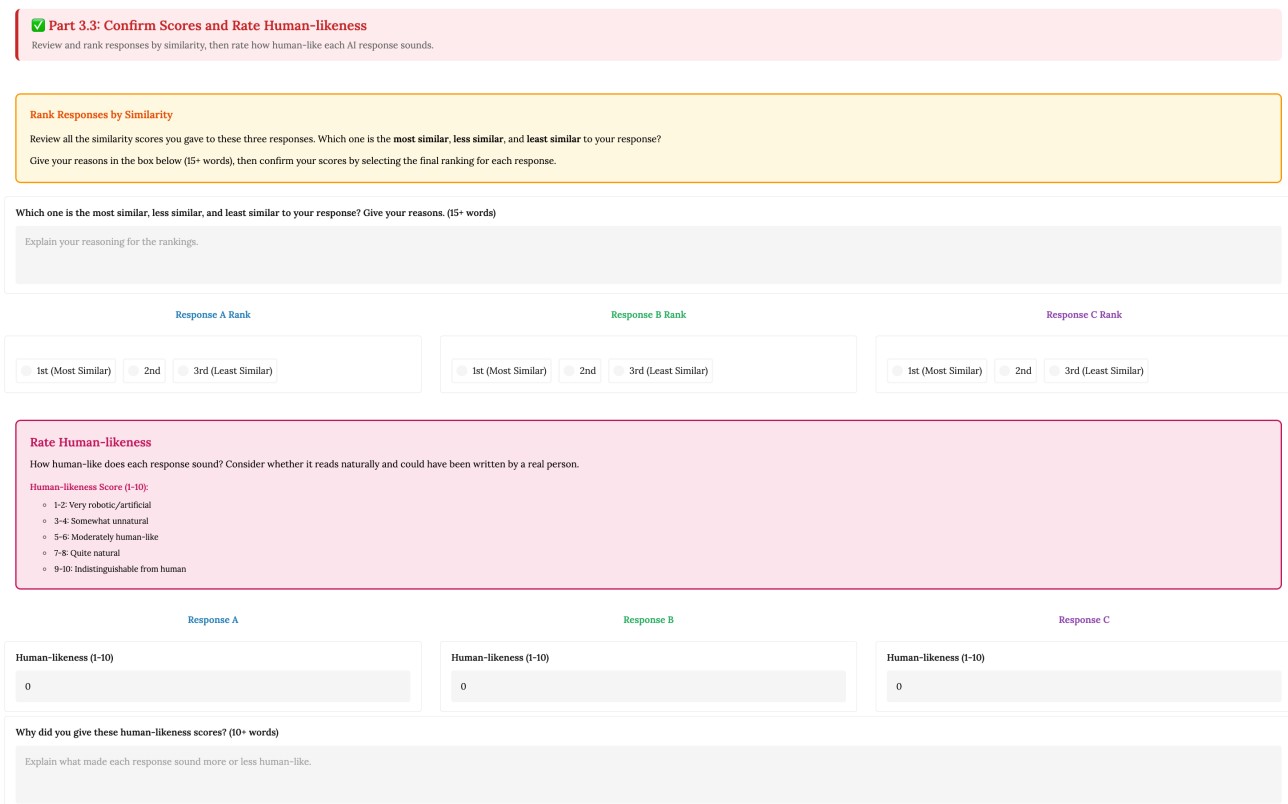

✅ **Part 3.3: Confirm Scores and Rate Human-likeness**
Review and rank responses by similarity, then rate how human-like each AI response sounds.

**Rank Responses by Similarity**

Review all the similarity scores you gave to these three responses. Which one is the **most similar**, **less similar**, and **least similar** to your response?

Give your reasons in the box below (15+ words), then confirm your scores by selecting the final ranking for each response.

Which one is the most similar, less similar, and least similar to your response? Give your reasons. (15+ words)

Explain your reasoning for the rankings.

| Response A Rank | Response B Rank | Response C Rank |
|---|---|---|
| ○ 1st (Most Similar)  ○ 2nd  ○ 3rd (Least Similar) | ○ 1st (Most Similar)  ○ 2nd  ○ 3rd (Least Similar) | ○ 1st (Most Similar)  ○ 2nd  ○ 3rd (Least Similar) |

**Rate Human-likeness**

How human-like does each response sound? Consider whether it reads naturally and could have been written by a real person.

Human-likeness Score (1-10):
  ○ 1-2: Very robotic/artificial
  ○ 3-4: Somewhat unnatural
  ○ 5-6: Moderately human-like
  ○ 7-8: Quite natural
  ○ 9-10: Indistinguishable from human

| Response A | Response B | Response C |
|---|---|---|
| Human-likeness (1-10) | Human-likeness (1-10) | Human-likeness (1-10) |
| 0 | 0 | 0 |

Why did you give these human-likeness scores? (10+ words)

Explain what made each response sound more or less human-like.

*Figure 18.* **Step 2.3 (Continue): Confirm rankings and humanlikeness evaluation.** Participants rank responses by similarity and rate how human-like each AI-generated response sounds, providing qualitative justifications.

