# OpenReview forum: "HumanLM: Simulating Users with State Alignment Beats Response Imitation"
_ICML.cc/2026/Conference — ICML 2026 regular_

### Official Review · Reviewer_rPbN · 2026-03-11

**Soundness:** 4
**Presentation:** 4
**Significance:** 4
**Originality:** 4
**Overall Recommendation:** 6
**Confidence:** 3

**Summary:**

This paper introduces a novel and reliable framework called HumanLM, which builds user simulators that precisely reflect real users’ current states by generating users’ latent states across six psychologically grounded state dimensions. HumanLM then synthesizes these aligned latent states into responses that represent real users, using GRPO training. For evaluation, the paper proposes a new benchmark, Humanual, which consists of six large-scale datasets with 26K users and 216K responses in total, spanning diverse tasks. Through extensive human evaluation, HumanLM achieves the highest score in terms of human-likeness.

**Compliance With Llm Reviewing Policy:**

Affirmed.

**Key Questions For Authors:**

- What would happen if a weaker or smaller LLM were used as the LLM-as-a-Judge during training? Would the results change significantly?

- In Figure 8, the average score of HumanLM achieves the best performance on both similarity and human-likeness. However, I wonder whether the authors conducted statistical significance tests for these improvements.

- In the training and evaluation split, to what extent do the user profiles differ between the two sets? (about "user profile generalizability")

- The following questions are meant to better understand the proposed method:
(1) Could the authors clarify the final reward formulation used in training?
(2) Did the authors observe any reward-hacking phenomena in this domain? For example, given certain states, does HumanLM tend to unconditionally generate specific responses, or prefer certain states regardless of the provided context or user profile? Could such behavior be one of the bottlenecks that makes simulating realistic user responses challenging?
(3) In Figure 4, the pattern of state alignment scores appears to vary across domains. For instance, in Humanual-Book, the communication score is noticeably lower than other state dimensions. Could the authors explain why this occurs, or provide further discussion about different patterns observed across domains?

**Strengths And Weaknesses:**

- The paper is very well written; the motivation is realistic and clearly applicable to real-world user-specific applications.

- The paper introduces a novel framework and benchmark for building robust and reliable user simulators, which makes a meaningful contribution to this research area.

- To demonstrate the effectiveness of the proposed framework, the authors conduct extensive experiments. For example, in Table 1 they compare HumanLM with different training techniques. In Section 6, which is particularly important, the authors further evaluate whether the proposed framework is effective for 37 real-world users.

- To ensure reliable results, the authors also conduct several additional and helpful analyses. For example, Figure 5 compares training dynamics, Figure 6 evaluates the reliability of the LLM-as-a-judge evaluation, and Figure 7 analyzes which dimensions are most important.

---

> ### Author Rebuttal · Authors · 2026-03-31
>
> We sincerely appreciate the reviewer’s approval! We address your questions:
>
> ---
>
> ## **[Question 1] "What would happen if a weaker LLM were used as the judge during training?"**
>
> We train a model on Humanual-Opinion using a `gpt-5.4-nano` judge (a weaker LLM)  and compare it with our model trained from a `gpt-5-mini` judge. The response alignment scores below are evaluated by `claude-4.5-haiku`.
>
> | | Belief | Goal | Value | Stance | Emotion | Communication | Response |
> |---|---|---|---|---|---|---|---|
> | HumanLM (gpt-5.4-nano) | 0.356 | 0.400 | 0.459 | 0.490 | 0.281 | 0.209 | 0.254 |
> | HumanLM (gpt-5-mini)| 0.269 |0.397| 0.469|0.499|0.291|0.204 | 0.256|
>
> Interestingly, the response alignment scores are similar, while `gpt-5-mini` is slightly better than `gpt-5.4-nano` on three state dimensions. We added this result to `Appendix D.3` to support the robustness of HumanLM training.
>
> ---
>
> ## **[Question 2] "statistical significance tests for (user study) improvement"**
>
> We wanted to update that we extended our online study from 37 participants to 111 (3x) participants for a broader population. We have modified our draft. Please view the figure [[link]]( https://pockiepockie428.github.io/HUMANUAL/user_study_results.pdf ) and the new results with statistical significance:
>
> “”“
>
> **For overall similarity scores**, HumanLM achieves the highest average score of 6.5 with a win rate of 41.4%. In contrast, `Qwen3-8b-think` and `GRPO-think` arrive at win rates of 30.6% (-10.8%) and 27.9% (-13.5%), respectively. 68.6% of the participants reported that HumanLM responses are “most similar” or “nearly identical” to theirs.
>
> **Statistical significance.** We conduct paired one-sided Wilcoxon signed-rank tests across the scores from 111 participants, confirming that HumanLM significantly outperforms both `Qwen3-8b-think` (p = 0.0279 < 0.05) and `GRPO-think` (p = 0.00284 < 0.01).
>
> **Humanlikeness scores**. 76.6% of the participants reported that HumanLM responses are “quite natural” or “indistinguishable from humans” , while only 72.1% reported the same for `Qwen3-8b-think`.
>
> ”“”
>
> We hope this further strengthens our results!
>
> ---
>
> ## **[Question 3] "to what extent do the user profiles differ between the two sets? "**
>
> We split the train/val/test set by post ids. Therefore, training and test sets have overlapping user profiles, while all the contexts are unseen during testing.
>
> ---
>
>
> ## **[Comment 4.1] "Could the authors clarify the final reward formulation used in training?"**
>
> Yes, for HumanLM, we train the model to directly generate one of the six state dimensions, which is scored by an LLM judge that evaluates if, for example, the generated stance (strongly disagree), aligns the stance in the ground truth. Meanwhile, when the model outputs a response, it firstly generates a reasoning trace with all state dimensions and then a final response. Similarly, we score the final response based on if it aligns with the ground truth. In short, the model is trained to **generate all state dimensions and synthesize responses** during training.
>
> ---
>
> ## **[Comment 4.2] " Did the authors observe any reward-hacking phenomena?"**
>
> Great question!
>
> Interestingly, we found the **reward hacking issue happens less in HumanLM**, since we have denser alignment signals. A reward hacking issue when training the standard GRPO baseline is that, in a multiturn user thread, the model can achieve high reward when it simply repeats exactly what the last user said. The reward hacking is resulted from users with similar opinions tend to discuss under the same thread, and the model can get very high reward by repeating. HumanLM, instead, prevents this behavior by more fine-grained alignment.
>
> A minor reward hacking issue that happened to HumanLM was that the length of latent states becomes longer, which is a common issue that can be prevented with length penalty.
>
> We have added the above discussion to our paper.
>
> ---
>
> ## **[Question 4.3] "In Figure 4, the pattern of state alignment scores appears to vary across domains... Could the authors explain why this occurs?"**
>
> Thanks for this question!
>
> One reason that the patterns are different is due to the variance of each state on different datasets. Basically, some states are easier to generate on some datasets while hard to generate on others. For example, on Humanual-News (YouTube), most users communicate with extremely short and aggressive sentences, while on other datasets, users leave comments with very different styles, making generating their communication styles much harder. This provides **good prior** to understand simulating users on target datasets.
>
> ---
>
> ## **Summary**
>
> We thank you for the extensive review and insights! We hope our answers address your questions on 1) using weaker LLM judge, 2) statistical significance tests, and 3) experimental details and analysis.
>
> Finally, we greatly appreciate your suggestions, which strengthens our claims in the paper. Thank you again!

---

> > ### Author Rebuttal · Reviewer_rPbN · 2026-04-03
> >
> > Thank you for addressing my questions.

---

> > > ### Author Response · Authors · 2026-04-08
> > >
> > > Thank you! We appreciate your time, insights, and support!
> > >
> > > Sincerely,
> > >
> > > Author of paper #2842

---

### Official Review · Reviewer_gpqT · 2026-03-11

**Soundness:** 4
**Presentation:** 4
**Significance:** 4
**Originality:** 4
**Overall Recommendation:** 5
**Confidence:** 3

**Summary:**

In this paper, the authors address existing limitations in user simulation and propose HumanLM, a novel training framework. This approach involves generating aligned latent states across six psychologically grounded dimensions (belief, goal, emotion, value, stance, and communication) and synthesizing these states into natural responses. The authors demonstrate that HumanLM significantly outperforms alternative approaches.

**Compliance With Llm Reviewing Policy:**

Affirmed.

**Final Justification:**

The authors addressed all of the minor issues raised. They supported the rebuttal with proper justifications, new results, and illustrations, thereby improving an already solid paper. Thus, I increased the rating in "Soundness."

**Key Questions For Authors:**

None

**Limitations:**

In the conclusion the authors acknowledge that future work can explore the diversity aspect of user simulator and multi-domain training. However, the paper would benefit from including some of the aforementioned minor issues as the limitations.

**Strengths And Weaknesses:**

Soundness.
The paper is technically sound. The method is based on well-established principles that are properly described and cited. The methodology is clearly formalized with well-defined equations. The six state dimensions introduced are grounded in psychology. The evaluation is also comprehensive; the authors compare HumanLM to several other methods using a new, specifically developed benchmark, Humanual.
- Minor issues:
1. A complexity and computational cost analysis would improve the paper, as it does not report the training times or costs of HumanLM compared to other methods.
2. The method relies heavily on LLM judges for both reward assignment during training and the final evaluation. The user study described in Appendix F seems limited.
3. There is no empirical validation showing that the introduced state dimensions accurately capture how real users generate responses. The authors assume that these dimensions are correct without testing whether they are exhaustive or psychologically realistic.
4. The section “5.2. Relations between States and Responses” explores how certain state dimensions contribute to responses. However, the work would benefit from a full-scale ablation study.

Presentation.
The submission is clearly written and well-structured. The “Related works” section properly positions the work in the context of prior studies. The paper provides sufficient detail to facilitate reproducibility.
- Minor issue:
The study uses several non-open-source LLMs: claude-4.5-haiku, gpt-5-mini, gemini-3-pro. This may hinder reproduction for some researchers and raise concerns about reliance on commercial APIs.

Significance.
This paper addresses the important issue of user simulation. Its scope of impact is quite broad. This approach has the potential to become the new standard in user simulation. Another important point is the specifically developed benchmark, Humanual, which can be used to validate a wide variety of user simulators.

Originality.
The work introduces a new method, as well as a specifically designed benchmark for its validation.

---

> ### Author Rebuttal · Authors · 2026-03-31
>
> We appreciate the reviewers' extensive and thoughtful outputs! We'll address each comment below:
>
> ---
>
> ## **[Comment 1] "A complexity and computational cost analysis would improve the paper"**
>
> We added a computational cost analysis to `Appendix C`. Here is what we added:
>  ```
> GRPO training of Qwen3-8B was run on 8 NVIDIA A100 80GB GPUs for 9.1 hours, totaling 72.8 GPU-hours for a full 276-step run. The final model was selected from checkpoint step 175.
>
> On Humanual-Opinion, GPT-5-mini judge usage during training cost approximately $2.63 per 25 steps, corresponding to about $18.41 up to the selected checkpoint at step 175. Validation during training cost approximately $5.37 per validation pass, or $37.59 total up to step 175. Final test evaluation with Claude Haiku 4.5 cost $45.20.
>
> The other Humanual datasets have similar training costs, while their validation cost was lower since the validation set sizes are smaller. We hope this provides extra information to understand our cost.
> ```
> ---
>
> ## **[Comment 2] "The method relies heavily on LLM judges for both reward assignment during training and the final evaluation. The user study seems limited."**
>
> We would like to clarify the following:
>
> - First, we deliberately use different LLM judges for training (`gpt-5-mini`) and evaluation (`claude-4.5-haiku`) which reduces the risk of overfitting to one judge.
>
> - Second, we also show that HumanLM and our baselines are ranked consistently across both `claude-4.5-haiku` and `gemini-3-pro` (`Figure 6`).
>
> - Third, we report additional validation beyond judge-based metrics. With embedding similarity HumanLM outperforms (`Appendix D.1`).
>
> - Our real-time user study validates our main results, and we have **updated our study to include 111 participants**. HumanLM achieves the highest similarity win rate (41.4%) and humanlikeness scores with statistically significant improvements (p < 0.01). For the details of user study, please refer to figure [[link]]( https://pockiepockie428.github.io/HUMANUAL/user_study_results.pdf) and our response to `Reviewer #4 rPbN [Question 2]`.
>
> We hope our clarifications on our evaluation design, additional metrics, and user study, can alleviate your concern on our eval setup.
>
> ---
>
> ## **[Comment 3] "empirical validation showing that the state dimensions accurately capture how real users generate responses"**
>
> In our human study (N=111), participants self-reported their own stance, emotion, belief, value, goal, and communication style before seeing any AI output. HumanLM, trained with these six dimensions, was rated most similar to users' actual responses 41.4% of the time (p < 0.01). We also conduct **new ablation experiment** in our next response to confirm each dimension captures a meaningful signal (22.9 → 25.6).
>
>
> ---
>
> ## **[Comment 4] "how certain state dimensions contribute to responses…a full-scale ablation study"**
>
> To show the contribution of state dimensions, we conduct a study where we incrementally add one dimension each time during training time (since trying all combinations will lead to extensive cost). Here are results:
>
> | State Dimensions| Response Score |
> |---|---|
> | Stance| 22.9|
> | Stance + Emotion| 24.5|
> | Stance + Emotion + Belief| 24.2|
> | Stance + Emotion + Belief + Value| 25.3|
> | Stance + Emotion + Belief + Value + Goal| 24.3|
> | Stance + Emotion + Belief + Value + Goal + Communication | 25.6|
>
> Link to figure: [[here]](https://pockiepockie428.github.io/HUMANUAL/saturation_line_humanlm.pdf)
>
> The key observation is that the response alignment score generally improves and seems to gradually reach a plateau, with the best performance achieved with all dimensions present. We hope this experiment will justify our design choice and show that contribution of the state dimensions.
>
>
> ---
>
> ## **[Comment 5] "The study uses several non-open-source LLMs...reproduction for some researchers"**
>
> - In addition to using a close-source LLM judge, we also provide an additional evaluation that computes embedding similarity using an open-sourced embedding mode from Snowflake. Please refer to Table 3.
>
> - Moreover, we agree an open-source evaluation is more friendly to the open-source community. We are working on getting the results using `kimi-k2.5`, one of the best open-source models, as the LM judge for ensuring the reproducibility. We will include the results in our final version. Thank you for this suggestion.
>
> ## **Summary**
>
> We deeply appreciate your approval and your thoughtful comments. We hope our answers address concerns about 1) computational costs, 2) evaluation, and 3) state dimension contribution to final responses. Please let us know if you have more questions! Thank you again!
>
> ---
>
> ### Reference
>
> [1] Wang et al. (2022). Self-Consistency Improves Chain of Thought Reasoning in Language Models.
>
> [2] Farquhar et al. (2024). Detecting Hallucinations in Large Language Models Using Semantic Entropy. Nature.
>
> [3] Kadavath et al. (2022). Language Models (Mostly) Know What They Know.

---

> > ### Author Rebuttal · Reviewer_gpqT · 2026-04-03
> >
> > Thank you for your exhaustive response. I consider all issues resolved.

---

> > > ### Author Response · Authors · 2026-04-08
> > >
> > > Glad to know the issues are addressed. We appreciate your efforts and support! Thank you!
> > >
> > > Sincerely,
> > >
> > > Author of paper #2842

---

### Official Review · Reviewer_yhum · 2026-03-13

**Soundness:** 1
**Presentation:** 2
**Significance:** 3
**Originality:** 2
**Overall Recommendation:** 3
**Confidence:** 4

**Summary:**

This paper aims to address the challenge of training LLMs to simulate real-world users. The authors argue that existing methods primarily imitate surface level patterns (like presence of emojis), failing to capture the underlying psychological states that drive user responses. To address this limitation, the paper proposes HUMANLM, a training framework that aligns six latent psychological dimensions (belief, goal, value, stance, emotion, communication) between generated and ground truth responses using GRPO. For evaluation, the authors introduce their benchmark of 6 datasets (HUMANUAL). The benchmark contains approximately 26k users and 216k responses. Experiments show that HUMANLM outperforms baseline methods such as SFT and GRPO on a proposed response alignment score, which measures how well generated responses reflect the latent psychological states of the ground-truth responses using an LLM judge. In addition, the authors conduct a user study with 37 participants to assess the perceived similarity and human-likeness of generated responses.

**Compliance With Llm Reviewing Policy:**

Affirmed.

**Final Justification:**

Multiple comments and questions are left to be clarified in the paper/rebuttal, this seems like a rushed work and demands more polishing. Persona creation is a major aspect of the paper because that is what the elicited values have to be aligned for downstream RL. I like the idea and overall approach, but the paper lacks soundness, so I will retain my original score.
Edit after author response:
Thanks to the authors for the additional exps. Some of my questions were answered in the response, raising my score by 1.

**Key Questions For Authors:**

1. How exactly are the “response alignment scores” in Table 1 computed and aggregated?
2. Why were the six state dimensions (belief, goal, value, stance, emotion, communication) selected? Did the authors experiment with alternative sets of dimensions or conduct ablation studies to assess the importance of each dimension?
3. How robust is the persona extraction process to sarcasm, fictional narratives, or noisy user responses (e.g., in Reddit datasets)? Have the authors evaluated the accuracy of the generated personas?
4. Since personas are generated from the earliest responses while evaluation responses may occur much later, how does the model handle temporal changes in user attributes or preferences?

Evaluation Independence:
Do the authors have evaluation metrics that do not rely on LLM judges (e.g., human annotation of latent states or behavioral prediction accuracy)? Such evaluation could help validate that the model truly captures user states rather than optimizing for judge preferences.

**Limitations:**

The authors discuss privacy protections for dataset collection and the user study, including de-identification and informed consent procedures. However, the paper could further discuss methodological limitations such as the reliance on LLM judges for both training and evaluation, potential biases introduced by persona extraction models, and the challenges of accurately modeling dynamic user attributes over time.

**Strengths And Weaknesses:**

Strengths:
1. The motivation of the paper is strong. Simulating user behavior of real world users is an important problem.
2. The authors propose HUMANUAL, a benchmark of 26k users and their 216k responses from multiple real-world sources (e.g., news comments, book reviews, online discussions, email threads)
3. The paper is generally well structured and easy to follow. The figures, particularly Figures 1 and 2, effectively illustrate the conceptual difference between response imitation and latent-state alignment, making the proposed framework intuitive to understand.

Weakness:
Soundness:
First, the evaluation methodology relies heavily on LLM-based judges. The primary metric, "response alignment score," is produced by prompting an LLM judge to evaluate the similarity between generated and ground-truth responses . The same paradigm is also used during training to compute reinforcement learning rewards. Although the authors use different models for training and evaluation judges, both follow similar prompting paradigms, raising concerns about evaluation circularity. The paper would be stronger if it included evaluation metrics independent of LLM judges, such as performance on survey benchmarks, or other behavior datasets.

Second, the latent state framework assumes that responses can be decomposed into psychological states (belief, goal, value, stance, emotion, communication). While these dimensions are individually motivated by literature, the paper does not justify why this particular set of six dimensions is sufficient for modeling user responses. Since these dimensions are central to the method, the lack of empirical validation weakens the argument that they meaningfully capture user behavior. Is this the universal set of required dimensions? What's the ceiling? The choice of latent dimensions is very important here because it is the primary difference between GRPO and HUMANLM.

Third, the persona construction pipeline raises several concerns. User profiles are summarized from at most the earliest 20 responses using a small LLM (Claude Haiku). Prior works have also used Reddit Comments for training LLMs / VLMs for comment simulation, noted such issues and shown methods to clean the dataset and mitigate such challenges [1] This raises several potential issues:

On Reddit, r/AITA is a popular subreddit with a lot of sarcasm and even concerns of LARPing, the prompts do not seem to mitigate any of these. Especially smaller LLMs. The datasets span long temporal ranges (in some cases decades), yet personas are derived from the earliest responses. This implicitly assumes that user traits remain stable over time, it is challenging to believe that a persona from 1974 would be applicable in 2001 as datasets are sourced in this time period as given in Table 2.
- It is assumed that their psychological traits do not chage
- Even if this assumption is correct, their demographics like Location, Occupation, Marital/Family Status, Interests, will change over time
- Age, the authors in their prompt denormalize the age, but don't explain how this used during inference time, do they change the persona's age at the time of inference? It is not mentioned anywhere.

These factors reduce confidence that the constructed personas faithfully represent real users.

The theoretical formulation does not fully match the implemented training procedure. The paper defines an objective based on indicator functions over latent state sets extracted from responses (Equation 1). However, the actual implementation uses LLM judges to produce continuous alignment scores for batches of generated states and responses. The relationship between this scoring mechanism and the theoretical formulation is not clearly explained, creating a gap between the conceptual objective and the practical training signal.

The authors mention in related work that they avoid Social Simulation and Survey based approaches because "simplifies responses into discrete actions, which lack of rich information to train or evaluate models in understanding more fine-grained user thoughts" however this would mean that HumanLM outperforms baselines on Social Survey Simulation if it is trained from richer data, these results seem to be missing and their lack seems potentially dismissive. One could rather argue that social simulation and surveys report (1) aggregate scores, reducing the idiosyncratic effects (2) retest accuracies. This makes the simulations statistically significant and quantifies uncertainity which is a primary challenge in user simulation, this is especially problematic as the authors themselves mention "In fact, even the same user may not perfectly reproduce their own responses." Can the authors establish the retest accuracy or reproducibility, or quantify the uncertainity in latent psychological traits?

Finally, the user study is based on only 37 participants, and authors give no statistical measures to evaluate the significance of the user study, only relative improvements.

[1] Singh, Somesh, et al. "Teaching Human Behavior Improves Content Understanding Abilities Of LLMs." ICLR 2025.

---

> ### Author Rebuttal · Authors · 2026-03-31
>
> We sincerely appreciate the reviewer’s comments. We address your concerns below:
>
> ---
>
> ## **[Comment 1, 9] "Evaluation methodology relies heavily on LLM judges... user study is based on 37 participants"**
>
> We've increased the user study from 37 to 111 participants (3x) and added statistical significance measures. Please see [here]( https://pockiepockie428.github.io/HUMANUAL/user_study_results.pdf) and our response to `Reviewer #4 rPbN [Question 2]` for details. Note that the user study is independent of LLM judges.
>
> ---
>
> ## **[Q1] "How are the “response alignment scores” [...] computed?"**
>
> The response alignment score measures how well a generation matches the ground truth. We extract several key points from the ground truth, score each 0–1 by how well the generated response reflects it, and average into a coverage score C. A penalty P is subtracted for irrelevant, hallucinated, or overly long content, giving $max(0, min(1, C−P))$. Table 1 reports these scores averaged across all instances per experiment.
>
> ---
>
> ## **[Comment 2 & Q2] "the paper doesn't justify why this particular set of six dimensions is sufficient"**
>
> We incrementally add one state dimension at a time during training see [here](https://pockiepockie428.github.io/HUMANUAL/saturation_line_humanlm.pdf) and see response to `Reviewer #3 gpqT [Comment 4]`. The plateauing curve demonstrates comprehensive coverage with six states.
>
> ## **[Comment 3 & Q3] "How robust is the persona extraction process to sarcasm?"**
>
> The persona prompt (Appendix E) requires evidence-based extraction with quoted responses and separates communication style from values and beliefs as distinct fields, handling sarcasm as a stylistic pattern rather than literal belief. For example, a Humanual-Opinion user's sarcastic comments ("A kids-ONLY wedding") and sincere support ("I am also proud of you!") are both correctly captured under communication style (the first as "Playful sarcasm in suggesting solutions," the second as "Direct emotional validation").
>
> ---
>
> ## **[Comment 4 & Q4] "datasets span long temporal ranges...how does the model handle temporal changes in user attributes or preferences?"**
>
> We clarify below:
>
> - The summarizer LLM is provided with the user history timestamp to capture temporal drift. We've added this to the paper.
> - Personas can be dynamically updated acoording to downstream tasks, in any case, HumanLM can still be applied.
> - Please note that persona generation is not our primary contribution. We showed that HumanLM's state alignment and synthesis algorithm is effective regardless of persona construction.
> - We believe temporal user modeling is an important future direction and added to the discussion.
>
> ---
>
> ## **[Comment 5] The authors in their prompt denormalize the age"**
>
> We acknowledge the discrepancy. Our intent is only to infer a rough age group, and since explicit age signals are mostly limited to Humanual-Opinion, the practical impact is minimal. We'll clarify in the paper.
>
> ---
>
> ## **[Comment 6] "Theoretical formulation does not fully match the implemented training procedure"**
>
> Our **evaluation procedure** matches the theoretical formulation (per Question 1) for computing the difference of 2 sets. While our training procedure does not exactly match the evaluation objective (unlike GRPO), our method still outperforms GRPO by large margins. Matching the theoretical formulation in training is not strictly necessary for better performance. We elaborate in Section 3.4.
>
> ---
>
> ## **[Comment 7] "The difference between HumanLM and Social Survey Simulation"**
>
> - Firstly, our comment "simplifies responses into discrete actions" refers to dataset selection in Humanual, not a comparison with social survey simulation methods; social survey simulation remains an important task.
> - Moreover, most social survey simulation baselines use prompting or SFT-based approaches, which we compare against in our experiments.
> - Lastly, we believe social simulation datasets offer more certainty but may lack the rich signals like Humanual. For a general user foundation model, the optimal data composition from both types of datasets is an interesting direction to study.
>
> We're happy to discuss if anything remains unclear.
>
> ---
>
> ## **[Comment 8] "the uncertainty in latent psychological traits"**
>
> Since we are training to generate latent states in reasoning traces, uncertainty can be quantified through LLM token-level probability [1], entropy [2,3], or verbalized confidence [4]. We note this as a future direction in the conclusion. Thanks for this question!
>
> ---
>
> ## **Summary**
>
> We hope our responses address your concerns by clarifying eval metrics and user study, adding experiments on state dimensions, discussing persona construction, and clarifying statements on social simulation. We've revised the draft accordingly.
>
> We would appreciate your reconsideration of our work in light of our responses. Thank you for your valuable insights!
>
> ---
>
> # Reference
> See Reviewer #3 gpqT

---

> > ### Author Rebuttal · Reviewer_yhum · 2026-04-03
> >
> > [Comment 1, 9]: Resolved
> > [Q1, C4 & Q4, C5, C6]: Multiple comments and questions are left to be clarified in the paper, while I understand the 5000 char limit, this seems like a rushed work and demands more polishing.
> > [C2 & Q2]: The plateauing curve explains that the value of these 6 is incremental, and diminishing, however I have questioned the choice of these 6 from a larger possible set.
> > [Comment 3 & Q3]: With sufficient experience in evaluating sarcasm, irony, etc I have empirically observed smaller LLMs like Haiku, GPT-4.1-Nano to be quite inaccurate, unless the authors have strong arguments against this, its hard to believe, and this is a major aspect because the profiling if inaccurate can lead to train the downstream model to rely on surface level cues due to the small noise.
> > [C7]
> > > For a general user foundation model, the optimal data composition from both types of datasets is an interesting direction to study
> >
> > While this is considerably better framing, it does not warrant ignoring comparison against these methods, I think these are still important baselines.
> >
> > Reviewer ePki also agrees with my comments on Persona Construction and challenges from Social Media which should not be left unattended, they are equally important for the soundness.

---

> > > ### Author Response · Authors · 2026-04-08
> > >
> > > Thanks `Reviewer #2 yhum` for engaging with us! Our original responses were 12k and largely reduced to fit within 5k. We apologize for the conciseness. We explain more here and add more experiments for your follow-ups.
> > >
> > >
> > > **[C2 & Q2] “the choice of these 6 from a larger set”**
> > >
> > > We compared our 6-dimension experiment with a 10-dimension experiment on Humanual-Opinion, adding 4 additional latent states (intent, preference, knowledge, confidence). The goal is to justify that the 6 dimensions cover broad aspects and are sufficient.
> > >
> > > |# latent dimensions|6|10|
> > > |:--|:--:|:--:|
> > > |Alignment score|25.6|22.8|
> > >
> > > Using 10 dimensions performs worse. We found the 4 additional latent dimensions have large overlap with the original 6 latent dimensions, causing instability during training, where optimizing for overlapping dimensions pulls the model in conflicting directions.
> > >
> > > **[C3 & Q3] “smaller LLMs are quite inaccurate in evaluating sarcasm”**
> > >
> > > Thanks for this insight! To evaluate how well small LLMs identify sarcasm, we apply claude-4.5-haiku on the FigLang Sarcasm Detection Shared Task (Ghosh et al., ACL 2020), with 1800 Reddit testing conversations with sarcastic / non-sarcastic labels.
> > >
> > > We few-shot prompted with 20 training examples (similar to our setup) and reported the F1 score compared to their leaderboard.
> > >
> > > |Rank|Method|F1|
> > > |:--:|:--|:--:|
> > > |13|Dual LSTM w/ hierarchical attn|0.600|
> > > |4|**claude-4.5-haiku**|**0.736**|
> > > |3|BERT+Local Context|0.737|
> > > |2|RoBERTa|0.750|
> > > |1|BERT+BiLSTM+NeXtVLAD+Ensemble+Augmentation|0.834|
> > >
> > >
> > > claude-4.5-haiku (20251001) achieves a relatively high accuracy of 73.6%, while all other higher-ranking models were finetuned on 4,400 training examples. We added this result to our Appendix D. This experiment informs an important direction to improve user profiles, as mentioned by the reviewer. Meanwhile, we hope it mitigates the concern about identifying sarcasm. Thanks!
> > >
> > > **[C7] “comparison against (social survey) baselines methods”**
> > >
> > > We train and evaluate HumanLM on SocSci210 (Kolluri et al., EMNLP 2025), a benchmark from 210 social science experiments. The task is to predict a user’s response to a certain stimuli given their demographic on a Likert scale. We train HumanLM for ~400 steps using LLaMA3-8B-Instruct::
> > >
> > > |Method|Acc|
> > > |:--|:--:|
> > > |LLaMA3-8B-Instruct Base|70.3|
> > > |+ SFT|69.1|
> > > |+ DPO|72.6|
> > > |**HumanLM**|**78.3**|
> > >
> > > HumanLM achieves SoTA performance, validating the effectiveness of HumanLM training methods on downstream social survey predictions which have ground-truth metrics. We added this to our experiment section which hopefully provides strong support to our claim.
> > >
> > > For the unclarified questions:
> > >
> > > **[C4 & Q4] "handle temporal changes in user attributes"**
> > >
> > > - To clarify, when we summarize the persona from user history, we provide the summarizer LLM with the timestamp, which emphasizes more recent comments.
> > > - In real applications, the persona can always be dynamically summarized from the most recent histories to incorporate temporal changes, where HumanLM can still apply.
> > > - Lastly, temporal modeling of users is important. Yet, it requires more effort orthogonal to this work. For example, training neural networks and time series modeling are usually studied in separate topics, which complement each other. We are sure that future work with temporal modeling will be a good addition.
> > >
> > >
> > > **[C5] "how age is used during inference"**
> > >
> > > The age info is mostly only available on Humanual-Opinion, with little impact on the other datasets. From the reasoning traces, we see the model may make analysis using the age information, such as "`a 70-year-old person who advocates for age limits in government`", and outputs a plausible response that fits the background.
> > >
> > >
> > > **[Q1, C6] "theoretical formulation v.s. training procedure"**
> > >
> > > - [Q1] "How are response alignment scores computed and aggregated?"
> > > Our formulation minimizes the **difference of two state sets** from the generation and ground truth. Therefore, to compute the response alignment score, we firstly extract key points from the ground truth, and each key point is scored 0-1 based on how well the generated response reflects it. These scores are averaged into a coverage score $C$. A penalty $P$ is then subtracted for irrelevant key points from the generation. The final score is $max(0, min(1, C − P))$, clamped to [0, 1]. These scores are averaged across all examples.
> > >
> > > - Indeed, HumanLM training that decomposes responses into state dimensions does not exactly match this evaluation objective. However, matching the theoretical formulation in the training is **not necessarily an advantage** or requirement. In fact, the Standard GRPO baseline that uses response alignment scores as rewards underperforms GRPO by large margins, which shows the **effectiveness** of our approach.
> > >
> > >
> > > **Summary.** Given the above, any reconsideration of our work would be genuinely appreciated. Please refer to our response to Reviewer #1 for a summary table. Thank you sincerely.

---

### Official Review · Reviewer_ePki · 2026-03-16

**Soundness:** 1
**Presentation:** 2
**Significance:** 1
**Originality:** 2
**Overall Recommendation:** 2
**Confidence:** 4

**Summary:**

This paper discusses training of large language models to simulate real-world users. The authors argue that existing approaches primarily imitate surface-level linguistic patterns in user responses and therefore fail to capture the underlying psychological states that drive user behavior. To address this, the paper proposes HUMANLM. The method introduces six psychological dimensions (belief, goal, value, stance, emotion, and communication) and trains a model to generate responses by first producing latent states aligned with those inferred from ground-truth responses. The model is optimized using reinforcement learning (GRPO) with an LLM-based judge that evaluates the alignment between generated states and reference responses.

For evaluation, the authors introduce HUMANUAL, a benchmark consisting of six datasets collected from multiple domains including social media discussions, product reviews, political blogs, email communication, and chat interactions. The benchmark contains approximately 26k users and 216k responses. The authors report that HUMANLM improves response alignment compared to baseline approaches such as supervised fine-tuning (SFT) and GRPO, where alignment is measured using an LLM-based scoring function. In addition, the paper presents a user study with 37 participants to evaluate the perceived similarity and human-likeness of the generated responses.

**Compliance With Llm Reviewing Policy:**

Affirmed.

**Final Justification:**

Having read the rebuttal and rebuttal's reply, I still maintain my position. While I believe in the plausibility of the idea, however, the empirical validation is too weak to justify an acceptance.

**Key Questions For Authors:**

1.⁠ ⁠Since personas are constructed from the earliest responses of each user, how do the authors account for potential changes in user attributes over time?
2.⁠ ⁠Why were the six specific latent dimensions chosen? Are these complete? Did the authors experiment with alternative dimensions or perform ablation studies to identify their importance?
3.⁠ ⁠⁠In datasets such as r/AITA, how does the persona extraction process alleviate noisy data from social media, and potential concerns of sarcasm.
4.⁠ ⁠⁠Have the authors considered comparing against stronger persona modeling methods or baselines such as retrieval/memory based persona conditioning
5. Downstream behavioral validity: Have you tested whether using HUMANLM as a simulator changes evaluations or choices in downstream tasks (e.g., policy rollout, product A/B testing) compared to prior simulators? If not, what is your plan to show that improved alignment yields better real-world outcomes?

**Limitations:**

No. While the authors include an impact statement discussing privacy and dataset sources, the paper does not sufficiently discuss methodological limitations such as reliance on LLM judges, potential bias in persona construction, and evaluation uncertainty. A clearer discussion of these limitations would strengthen the work.

**Strengths And Weaknesses:**

Strengths

Using latent states for generation and alignment: reframing simulation from imitating surface text to generating and aligning latent, psychologically-inspired state descriptions is a compelling idea and well motivated by sociolinguistic and cognitive theory.
HUMANUAL Benchmark: a comprehensive benchmark spanning 6 diverse datasets of real user responses with chronologically split test sets.
Empirical gains & analyses: the paper reports consistent gains on state and response alignment (avg. +16.3% relative), embedding similarity improvements, training dynamics visualizations, ablations and a small real-time human study (37 participants) confirming better perceived similarity and humanlikeness.
Method Design: The authors have addressed challenges and potential edge cases (such as combinatorial ambiguities) and designed a robust framework for user simulation.



Weaknesses

- Choice and coverage of state dimensions: the six chosen dimensions are plausible, but the mapping from free-form user responses to an exhaustive set of latent states is underspecified. It is unclear how the state set scales to complex, cultural, or domain-specific signals, and whether the dimensionalization induces bias (over/under-representing some aspects).

- Reliance on LLM judges as ground truth: both state and response alignment metrics depend heavily on LLM judges - improvements may reflect better optimization of LLM-judge scoring rather than true alignment to human latent states.

- Evaluation breadth: most reported metrics are LLM-judge alignment scores and embedding similarity; while helpful, they do not directly measure downstream behavioral validity (e.g., whether a policy tuned with these simulators yields the same real-user response distribution or decisions). The small online study is promising but limited in scope and population.

- several methodological issues raise concerns about the robustness of the results:
1.⁠ ⁠The central component of the method is the generation and alignment of latent psychological states, however there is no validation through annotations of these being true. An LLM as a Judge evaluates the inferred and target states, and another LLM measures their similarity. The only empirical validation is a user study of 37 participants which is not enough to prove the claims made by the paper.
2.⁠ ⁠⁠Persona construction also raises concerns. Personas are generated from the earliest 20 responses of each user. In datasets such as Reddit discussions (e.g., r/AITA), responses often contain sarcasm and satire. The paper does not discuss how such cases are handled. Since persona extraction relies on Claude-Haiku, a relatively small model, it is possible that ironic or narrative content may be interpreted as genuine user traits. Additionally, the use of the earliest responses assumes that user characteristics remain stable over time. However, the datasets span long time periods (3 to 25 years), and attributes such as interests, location, occupation, or even values change over time. The paper does not address this potential temporal drift. While the authors normalize the inferred age it is not mentioned how they use the age group during inference.
3.⁠ ⁠⁠Another concern is the selection of latent dimensions. While the six indivudual dimensions are motivated by prior literature, the paper does not empirically justify their choice.
4.⁠ ⁠⁠Finally, the baselines used in the experiments are relatively limited. Most baselines are variants of the same base model (Qwen3-8b) with different training procedures. The paper does not compare against stronger persona modeling approaches such as retrieval-augmented persona conditioning or memory-based user simulation methods.

Presentation
The paper is generally well structured and the conceptual motivation is clearly illustrated through figures and examples. However, some methodological details are missing or unclear. In particular, the evaluation metric and persona construction pipeline require more precise descriptions to allow readers to fully understand and reproduce the results. The figures are helpful and intuitive, especially those illustrating the latent state reasoning process.

Significance
User simulation is an interesting and important problem; The idea of modeling responses through intermediate psychological states is promising and may inspire further research on structured modeling of user behavior. However, the current work does not yet demonstrate clear advantages over simpler approaches due to uncertainties in evaluation and persona reliability.

Originality

Similar ideas have appeared in related work on persona modeling or behavior simulation using Theory of Mind approaches, the paper does not sufficiently distinguish from them but the explicit alignment of batch wise generated states with reference responses is an interesting choice. The training framework largely builds on existing reinforcement learning methods with LLM-based reward functions, so the methodological novelty is incremental.

The dataset contribution is valuable if correct justifications are made.

---

> ### Author Rebuttal · Authors · 2026-03-31
>
> Thank you for your thoughtful suggestions! Below we address your concerns.
>
> ---
>
> ## **[Comment 1, 4.3 & Q2] State dimensions**
>
> - **Why these six dimensions?** They derive from four psychological aspects in prior literature, forming a comprehensive set that captures how users respond and make impact.
>
> - **Are they complete?** While additional dimensions are plausible, these six generalize effectively across all datasets. Users can also add/remove dimensions, providing flexibility for HumanLM training.
>
> - **Scaling to complex/cultural/domain-specific signals?** Our datasets (e.g., from YouTube) are highly diverse, spanning multiple cultures and language styles. See examples at this [anonymous website](https://pockiepockie428.github.io/HUMANUAL/).
>
> - **Empirical justification?** We conduct an ablation study. Please see our response to `Reviewer #3 gpqT [Comment 4]` and the figure [[here]](https://pockiepockie428.github.io/HUMANUAL/saturation_line_humanlm.pdf). **Key finding**: alignment score plateaus after 3–5 dimensions, confirming the six dimensions are comprehensive.
>
> - **Contribution**: We'd like to note that state dimensions are not our primary contribution. HumanLM is a novel algorithm that performs state alignment and synthesis to accurately model responses, regardless of which dimensions are chosen.
>
> ---
>
> ## **[Comment 2] "Over-optimization of LLM-judge scoring?"**
>
> We re-train using a weaker judge (`gpt-5.4-nano`) and use `claude-4-5-haiku` for eval:
>
> ||Belief|Goal|Value|Stance|Emotion|Comm.|Response|
> |---|---|---|---|---|---|---|---|
> |HumanLM (gpt-5.4-nano)|0.356|0.400|0.459|0.490|0.281|0.209|0.254|
> |HumanLM (gpt-5-mini)|0.269|0.397|0.469|0.499|0.291|0.204|0.256|
>
> Scores are nearly identical, confirming gains are from better alignment, not higher-quality judge scores.
>
> ---
>
> ## **[Comment 3 & Q5] "Online study limited in scope"**
>
> - We expanded our study from 37 to **111 participants** (3×). See figure [[here]](https://pockiepockie428.github.io/HUMANUAL/user_study_results.pdf) and `Reviewer #4 rPbN [Question 2]` for details.
> - Policy tuning with user simulators is valuable but we believe it is currently out of scope, as it requires extensive resources beyond optimizing the simulator itself. We have added it as future work.
>
> ---
>
> ## **[Comment 4.1] "No annotation-based validation of latent state alignment"**
>
> - We also report embedding similarity scores `(Table 3, Appendix D)`, which require no LLM judge. Our 111-person user study doesn't depend on the judge as well.
> - Our evaluation results using claude and gemini yield consistent rankings, where we used `gpt-5-mini` as the LM judge in training to avoid evaluation bias.
> - LLM judge scores are grounded in real user responses. The judge directly compares generations against ground truth rather than subjectively predicting likelihoods, making it less prone to bias.
>
> ---
>
> ## **[Comment 4.2 & Q1, 3] "Personas from earliest 20 responses; datasets span..."**
>
> - We conduct human evaluation confirming that `claude-4.5-haiku` correctly interprets irony as style rather than literal belief (e.g., '*Absolutely brilliant plan I know*' →'*Employs sarcasm when mocking poor decisions*').
> - Also, we provide summarizer LLM with the timestamp of history, helping it weigh recent responses appropriately.
> - Our persona construction is **closer to real-world cases** than synthetic profiles with predefined demographics used in prior work.
> - **Inferred age** is one optional signal among many. Note that the model need not use it, but it can inform tone and cultural background (e.g., a 70-year-old critic of political figures).
>
> ---
>
> ## **[Comment 4.4 & Q4] "Baselines are limited"**
>
> We implement a RAG baseline retrieving the top-20 most relevant past comments as few-shot examples:
>
> |Model|Belif|Goal|Value|Stance|Emo|Comm|Response|
> |---|---|---|---|---|---|---|---|
> |Qwen3-8B|24.8|31.0|36.2|38.8|21.8|16.2|18.7|
> |Qwen3-8B+RAG|21.8|25.8|34.4|33.0|18.5|11.7|18.9|
> |HumanLM|26.9|39.7|46.9|49.9|29.1|20.4|25.6|
>
> RAG and persona baselines perform comparably, while HumanLM substantially outperforms both, confirming that gains come from state-alignment training.
>
> ---
>
> ## **[Comment 5] "Methodological details unclear"**
>
> We revised Section 4 with clearer descriptions of user profile construction and evaluation metrics.
>
> ---
>
> ## **[Comment 7] "Methodological novelty"**
>
> Prior work has explored RL and LLM reward functions, designing them for user simulation was an open question. Unlike previous works that rely on prompting or SFT, HumanLM targets latent state simulation rather than exact expressions. This paradigm shift is non-trivial. Moreover, state alignment provides a rich learning signal for noisy user data, and response synthesis ensures outputs adhere to the generated states.
>
> ---
>
> ## **Summary**
>
> We provide extensive clarifications, ablations on state dimensions, a RAG baseline, and more. We would sincerely appreciate your reconsideration and welcome any remaining concerns.

---

> > ### Author Rebuttal · Reviewer_ePki · 2026-04-04
> >
> > Thank you for providing detailed rebuttal. I truly appreciate it.
> > It is hard to address the methodological limitations that were pointed out in the review in the limited rebuttal space. Further, I still hold that the metrics used to judge the study do not directly measure downstream behavioral validity (whether it is 37 people or 111).

---

> > > ### Author Response · Authors · 2026-04-08
> > >
> > > Thank you for the follow-up! We carefully clarify several points in details:
> > >
> > > **[Comment 7] “the methodological novelty”**
> > >
> > > While reinforcement learning methods and LLM-based reward functions are both explored by previous works, how to design them for simulating from user data is unclear before our approach. As we discussed in the Related Work section, most of the Theory of Mind (pointed out by the reviewer) and social simulation works conduct prompting or SFT. HumanLM instead focuses on a more real-world objective that simulates latent states instead of the exact expression. We believe the paradigm transformation here is not trivial.
> > >
> > > Moreover, our framework aligns models with the latent states, ensuring a rich signal for the model to learn from “noisy” and seemingly “random” user data. Without this latent state alignment, the simulation performance is much weaker, as compared with the Standard GRPO baseline. Finally, our response synthesis process makes sure the final response adheres to the generated latent states.
> > >
> > > ---
> > >
> > > **[Further comment] “downstream behavior validity”**
> > >
> > > Downstream behavior validity (how policies or which product design should be chosen) is an important direction of user simulation, which is definitely of high interest and relevance in user simulation fields. Yet, having this behavior validity evaluation is difficult due to the requirement of knowing the population we are simulating and validating the behavior with the same population.
> > >
> > > - **For policy rollout**: An experiment can be training policy models with regard to our user simulator’s feedback, and evaluate real user’s preference to see if the policy model improved after training. Or, selecting two policy models and studying the correlation of preference from user simulators and real users. Either case, since simulation is per user, this requires knowing the real users and collecting their data for training their simulators, which is not applicable for anonymous users on the internet.
> > > - **For product A/B testing**: Similarly, collecting real user data requires a much larger scale of HCI study that goes beyond the scope of our work.
> > >
> > > Therefore, we clarify that the goal of this work is to establish a training framework via latent state alignment, rather than evaluating full end-to-end downstream applications, which require task specific infra, large scale experiments, and human resources. Within the scope of this work, we provide multiple complementary validations, which are consistent and commonly used evaluation methods in recent works:
> > >
> > > - **Human evaluation**
> > > - **Embedding similarity (LLM judge-independent)**
> > > - **Cross-judge robustness**: Different LLM judges for training achieve similar results and different LLM judges for testing produce consistent rankings.
> > >
> > > The first study is in line with validating user behaviors in terms of collecting opinions to daily life matters. We understand these evaluations don’t directly measure downstream behavioral effects. Similar to many works that focus on improving modelling or training framework, we view this downstream validation as a subsequent step.
> > >
> > > ---
> > >
> > > # **Summary**
> > >
> > > We summarize our response to you and other reviewers below:
> > >
> > >
> > > ||Reviewer #1|Reviewer #2|Reviewer #3|Reviewer #4|Action|
> > > |:---|:---|:---|:---|:---|:---|
> > > |**Presentation**|“well-structured, some details unclear”|“well-structured and easy to follow”|“clearly written and well-structured”|"Presentation: excellent."|Revised paper with clearer descriptions of user profile construction etc.|
> > > |**Motivation**|"interesting and important"|“motivation is strong.”|"addresses the important issue… impact is quite broad"|"Significance: excellent"|NA|
> > > |**Originality**|"unclear compared to persona / ToM work."|“fair”|“excellent”|"excellent"|Clarified our novelty in this response|
> > > |**Evaluation**|evaluation relies on LLM judges and lacks downstream behavioral validity|evaluation relies on LLM judges|"good-to-excellent evaluation"|Extensive experiments and analyses viewed positively|  (1) Expanded user study (**37 → 111 participants**) (2) clarified our consistent results across different LLM judges (3) clarified that embedding-similarity metrics in our paper is independent of LLM judges (4) trained HumanLM with a weaker judge model and obtained similar performance|
> > > |**Effectiveness**|“consistent gains”|asked statistical significance of the user study|”significantly outperforms baselines”|”extensive”|Added statistical significance for our user study|
> > > |**In-depth experimental study**|if the 6 dimensions are sufficient|Same as reviewer #1|a full-scale ablation study on state dimensions|NA|Added ablation to justify the 6 state dimensions|
> > > |**Baselines**|comparison against RAG|comparsion with social survey simulation methods|NA|NA|Added a RAG baseline and evaluated HumanLM on a social survey simulation benchmark|
> > > |**Reviewer response**|Partial resolvement|Partial resolvement|Fully addressed|Fully addressed| |
> > >
> > >
> > > Thank you.

---

### Decision · Program_Chairs · 2026-04-30

**Decision:**

Accept (regular)

**Comment:**

The paper addresses human simulation with latent natural language latent states (six aspects: belief, goal, value, stance, emotion, communication). Essentially, it's a type of chain-of-thoughts for human simulation. The paper further develops a dataset containing 26K users and 216K responses.

Reviewers generally recognize that the motivation is strong and the dataset is helpful. However reviewers also raised a few important concerns:
1. The choice of the six aspects (belief, goal, value, stance, emotion, communication) does not appear systematic enough.
2. The evaluation heavily replies on LLM-as-judge.